# Quantitative 3D histochemistry reveals region-specific amyloid-β reduction by the antidiabetic drug netoglitazone

Francesca Catto[1,2☯], Ehsan Dadgar-Kiani[3☯], Daniel Kirschenbaum[1☯],
Athena E. Economides[1], Anna Maria Reuss[1], Chiara Trevisan[1], Davide Caredio[1],
Delic Mirzet[1], Lukas Frick[1], Ulrike Weber-Stadlbauer[4,5], Sergey Litvinov[6],
Petros Koumoutsakos[6], Jin Hyung Lee[3,7,8,9‡*], Adriano Aguzzi[10‡*]

**1** Institute of Neuropathology, University Hospital Zurich, University of Zurich, Zurich, Switzerland, **2** IMAI MedTech GmbH, Zurich, Switzerland, **3** Department of Bioengineering, Stanford University, Stanford, California, United States of America, **4** Institute of Veterinary Pharmacology and Toxicology, University of Zurich, Zürich, Switzerland, **5** Neuroscience Center Zurich, University of Zurich and ETH Zurich, Zurich, Switzerland, **6** Computational Science and Engineering Laboratory, School of Engineering and Applied Sciences, Harvard University, Cambridge, United States of America, **7** Department of Neurology and Neurological Sciences, Stanford University, California, United States of America, **8** Department of Electrical Engineering, Stanford University, Stanford, California, United States of America, **9** Department of Neurosurgery, Stanford University, Stanford, California, United States of America, **10** Institute for the Science of the Aging Brain, St. Gallen, Switzerland

☯ These authors contributed equally to this work.
‡ JHL and AA also contributed equally to this work.
* adriano.aguzzi@isab.science (AA); ljinhy@stanford.edu (JHL)

## Abstract

A hallmark of Alzheimer's disease (AD) is the extracellular aggregation of toxic amyloid-beta (Aβ) peptides in form of plaques. Here, we identify netoglitazone, an antidiabetic compound previously tested in humans, as an Aβ aggregation antagonist. Netoglitazone improved cognition and reduced microglia activity in a mouse model of AD. Using quantitative whole-brain three-dimensional histology (Q3D), we precisely identified brain regions where netoglitazone reduced the number and size of Aβ plaques. We demonstrate the utility of Q3D in preclinical drug evaluation for AD by providing a high-resolution brain-wide view of drug efficacy. Applying Q3D has the potential to improve pre-clinical drug evaluation by providing information that can help identify mechanisms leading to brain region-specific drug efficacy.

## Introduction

Alzheimer's disease (AD) is a prevalent neurodegenerative disease which causes an inexorable decline in cognitive abilities, affecting the life of patients and of their caregivers and eventually leading to dementia and death [1]. Given that the strongest risk factor for AD is age, and considering that life expectancy is increasing in most parts of the world, it is anticipated that the incidence of AD will increase.

**Data availability statement:** The data underlying the results presented in the study are available from: https://doi.org/10.5281/zenodo.11121432

**Funding:** Adriano Aguzzi, A.A., is supported by institutional core funding by the University of Zurich and the University Hospital of Zurich, a Distinguished Scientist Award of the NOMIS Foundation (https://nomisfoundation.ch/), and grants from the GELU Foundation, the Swiss National Science Foundation (https://www.snf.ch/en) (SNSF grant ID 179040 and grant ID 207872, Sinergia grant ID 183563), the Human Frontiers Science Program (https://www.hfsp.org/) (grant ID RGP0001/2022), and the Michael J. Fox Foundation (https://www.michaeljfox.org/) (grant ID MJFF-022156). Jin Hyung Lee, JHL, is supported by the following funding: NIH/NINDS DP1 NS116783 NIH/NINDS R01 AG064051 NIH/NINDS R01 EB030884 Sergey Litvinov, SL, is supported by the European High Performance Computing Joint Undertaking (EuroHPC) Grant DComEX (956201-H2020-JTI-EuroHPC-2019-1). The funders played a role in the study design, data analysis, decision to publish, and preparation of the manuscript.

**Competing interests:** The authors have declared that no competing interests exist.

In AD, the amyloidogenic peptide amyloid beta (Aβ) is liberated from its precursor protein APP and aggregates into fibrils, giving rise to structures termed "neuritic plaques" by Alois Alzheimer. These aggregated species serve as templates and seeds for the nucleation of further Aβ [2–4] aggregation. According to the amyloid cascade hypothesis [5] Aβ aggregation is the primary cause of AD which induces all downstream aspects of neurodegeneration (aggregation of Tau protein, astrocyte and microglia activation, and eventually neuronal loss). Amyloid plaques have long been recognized as a pathological hallmark of AD and remain central to understanding disease progression and therapeutic strategies. However, studies indicate that soluble Aβ oligomers are the relevant neurotoxic agents in AD. Nevertheless, plaques contribute to localized tissue damage and neuroinflammation, and, also act as reservoirs for toxic oligomers influencing their bioavailability. This underscores the complexity of plaques in AD pathology, highlighting their interplay with soluble Aβ oligomers as both drivers and modulators of disease mechanisms [6,7]. While familial forms of AD are often caused by Aβ overexpression, there is ongoing debate about the importance of Aβ aggregation in sporadic AD, which constitutes the majority of cases [8]. Many studies have demonstrated that total Aβ deposition in the human brain correlates poorly with cognitive decline [9], and intellectually healthy individuals can harbor substantial Aβ plaque loads without displaying symptoms of dementia. This disconnect between plaque burden and cognitive decline raises important questions about the role of Aβ in disease progression.

The challenges in linking plaques to cognitive outcomes are further highlighted by the mixed results of therapeutic trials targeting Aβ. Anti-Aβ antibodies, such as crenezumab, aducanumab, solanezumab, lecanemab, and gantenerumab, have shown impressive efficacy in clearing brain Aβ but have delivered only marginal improvements in clinical outcomes [10–12]. Additionally, these treatments have been associated with severe side effects, including amyloid-related imaging abnormalities (ARIA) such as brain edema and bleeding [13,14]. The combination of modest clinical efficacy, significant adverse effects, prolonged treatment duration, and high costs underscores the need for a deeper understanding of AD pathology and the development of more effective and targeted therapeutic strategies.

While none of the clinical trials with anti-Aβ antibodies have yet delivered substantial therapeutic results, it is interesting to note that some antibodies had no discernible effect on the course of the disease whereas others appear to yield statistically meaningful, though very limited, beneficial effects [15–18]. The failure rate of AD clinical trials may be attributed to multiple reasons. Each of these antibodies target different epitopes and even different aggregational states and conformers of Aβ, and such parameters are likely to influence their therapeutic efficacy. However, the expression of APP and of the components responsible for its catabolic conversion into Aβ (e.g., BACE-1, presenilins, nicastrin, and several other proteins) is known to vary by anatomical brain region [19–21]. This would result in distinct region-specific drug efficacies. In addition, many of the brain functions affected by AD are related to specific brain regions, not necessarily overlapping with the regiospecificity of AD-drugs [22,23]. However, the available data on the regional distribution of Aβ in

the brain are limited, mostly due to lack of suitable technology. Amyloid load can be measured in clinical and preclinical studies using various techniques, including PET imaging, stereometric immunohistochemistry, ELISA, and western blotting [24–28]. However, these approaches either lack scalability to sample the entire brain (histology) or have no or very low spatial resolution (ELISA, PET). While modern PET systems for rodents achieve sub-millimeter resolution, which is sufficient for regional analyses, their resolution is still insufficient to discriminate individual amyloid plaques. Taking into account the highly compartmentalized structure of the brain and region-specific functions and symptoms, selective vulnerability, and pharmacodynamics [29] the testing of AD drugs requires imaging tools that are highly sensitive and can afford high spatial resolution.

Investigation of cheaper and less invasive therapies, such as lifestyle interventions, non-drug interventions, and repurposed drugs, may provide alternative approaches that can complement or replace current treatments for AD [30]. In addition, repurposing existing drugs already approved for other diseases may provide a more efficient and cost-effective approach to developing new treatments for AD [31].The present work builds on a previous study that screened compounds to target Aβ aggregation in vitro and identified netoglitazone, an FDA-approved thiazolidinedione (TZD) family antidiabetic compound, as an Aβ modifier from *in vitro* [32]. Here, we present a standardized procedure for screening anti-amyloid compounds in vivo. Our pipeline includes high-resolution 3D pharmacodynamic analysis, RNA sequencing, and behavioral assays to test molecules at the brain level. The in vivo tests show that netoglitazone reduces Aβ load and microglia activity in a region-specific manner and improves cognition in Alzheimer's mouse models. Our approach is generalizable and applicable to any anti-Aβ compound.

## Materials and methods

### APPS1 mice

APPPS1 transgenic mice were used in the study, which co-express the Swedish mutation K670M/N671L and PS1 mutation L166P under the control of the neuron-specific Thy-1 promoter on a C57BL/6 genetic background [33]. In our breeding scheme, 68% of the APPPS1 transgene-carrying parents were male, while 32% were female. APPPS1 mice were habituated ahead of the study to voluntarily drink condensed milk formulation from a pipette. The condensed milk used in the study is commercially available (Migros) and contains milk, sugar, stabilizer E339. Body weight was measured ahead of commencing the study to calculate the dose of netoglitazone for each mouse and to calculate the total blood volume. Number and sex of mice per each experiment is detailed in Table 1.

### Animal treatments and tissue preparation

All animal experiments were carried out in strict accordance with the Rules and Regulations for the Protection of Animal Rights (Tierschutzgesetz and Tierschutzverordnung) of the Swiss Bundesamt für Lebensmittelsicherheit und Veterinärwesen and were pre-emptively approved by the Animal Welfare Committee of the Canton of Zürich (permit 040/2015). APPPS1 male and female mice were treated daily orally with netoglitazone (Wren Therapeutics, Cambridge UK) diluted in condensed milk (Migros, Switzerland) and PBS. The administered dosages were either 75mg/ml (high-dose) or 25mg/ml (low-dose). The treatment duration was either 90 or 180 days (short-term or long-term, respectively). The dose was selected based on previous work where pharmacokinetics showed that netoglitazone crossed the blood-brain barrier after oral administration (15mg/Kg) and could be detected in micro dialysate from fraction 30–60 min post administration [32]. Control mice were treated with PBS (PBS and condensed milk). The starting age of the treated mice were 56 ± 4 days (Fig 1A). For whole-brain analysis of Aβ plaques, the protocol was performed as previously described [22]. Briefly: after treatments were completed, mice were deeply anaesthetized with ketamine and xylazine and transcardially perfused first with ice cold PBS, followed by a hydrogel monomer mixture of 4% acrylamide,0.05% bisacrylamide, and 1% paraformaldehyde. Brains were harvested, post incubated in hydrogel mixture overnight and further cleared. For whole-hemisphere analysis of microglia, 2D immunofluorescence (IF), 2D immunohistochemistry

**Table 1. Number and sex of mice per experiment.**

| Figure | Description | Number of Mice |
|---|---|---|
| Fig 1B | **180 days of treatment with netoglitazone induces anxiolytic-like effects, alters fear memory and restores deficit in temporal order memory in AD mice.** | 11 Tg mice per group of which 6 females and 5 males. |
| Fig 1C | **180 days of treatment with netoglitazone induces anxiolytic-like effects, alters fear memory and restores deficit in temporal order memory in AD mice.** | 21 Tg high dose treated mice (9 males and 12 females) and 22 Tg placebo treated mice (13 females and 9 males). 8 WT high dose treated mice (4 females and 4 males) and 8 WT placebo treated mice (4 females and 4 males). |
| Fig 1D | **180 days of treatment with netoglitazone induces anxiolytic-like effects, alters fear memory and restores deficit in temporal order memory in AD mice.** | 9 Tg mice and 4 WT mice per group (all males). |
| Fig 1E | **180 days of treatment with netoglitazone induces anxiolytic-like effects, alters fear memory and restores deficit in temporal order memory in AD mice.** | 21 Tg high dose treated mice (12 females and 9 males) and 22 Tg placebo treated mice (12 females and 9 males) 8 WT high dose treated mice (4 females and 4 males) and 8 WT placebo treated mice (4 females and 4 males). |
| Fig 2 | **Voxel-based whole-brain analysis shows regional and dose-dependent effects of netoglitazone in decreasing plaque count.** | 11 high dose, 10 low dose, 12 placebo Tg mice (full brains) of which 17 females and 16 males. |
| Fig 3 | **Voxel-based whole-brain analysis shows regional and dose-dependent effects of netoglitazone in decreasing plaque mean size.** | 11 high dose, 10 low dose, 12 placebo Tg mice (full brains) of which 17 females and 16 males. |
| Fig 4 | **Voxel-based whole-hemisphere analysis shows regional and dose-dependent decrease of microglia.** | 3 mice per group (3 half brains) of which: - 2 females and 1 male for high dose and placebo treated mice. - 2 males and 1 female for the low dose treated mice. |
| Fig 5 | **Gene expression changes are revealed by RNAseq upon long-term treatment with netoglitazone.** | 7 mice high dose, 3 mice low dose, 7 mice placebo (half a brain per mouse) of which 10 females and 7 males. |
| S1 Fig | **180 days of treatment with netoglitazone does not improve spatial recognition memory and basal locomotor activity.** | 21 Tg high dose treated mice (9 males and 12 females) and 22 Tg placebo treated mice (13 females and 9 males). 8 WT high dose treated mice (4 females and 4 males) and 8 WT placebo treated mice (4 females and 4 males). |
| S2 Fig | **Raw data of Aβ plaques and microglia staining.** | Plaques: 11 high dose, 10 low dose, 12 placebo Tg mice (full brains) of which 17 females and 16 males. Microglia: 3 mice per group (3 half brains) of which: • 2 females and 1 male for high dose and placebo treated mice. • 2 males and 1 female for the low dose treated mice. |
| S3 Fig | **Average number and size of plaque in the different cohorts.** | Plaques: 11 high dose, 10 low dose, 12 placebo Tg mice (full brains) of which 17 females and 16 males. Microglia: |
| S4 Fig | **2D immunohistochemistry and immunofluorescence of Aβ plaques.** | 3 mice per group (half a brain per mouse) of which 5 females and 4 males. |
| S5 Fig | **Segmentation of microglia cells. Representative images depicting the quality of microglia segmentation.** | 3 mice per group (3 half brains) of which: • 2 females and 1 male for high dose and placebo treated mice. • 2 males and 1 female for the low dose treated mice. |

(IHC), and RNA sequencing (RNAseq) analysis: mice were deeply anaesthetized with ketamine and xylazine and transcardially perfused first with ice cold PBS, followed by 4% paraformaldehyde. Brains were harvested, and the hemispheres were separated. Left hemispheres were further incubated in the paraformaldehyde solution for 24 hours, then moved to 30% sucrose in PBS for two days at 4 °C and finally they were embedded in paraffin to be further used for whole-hemisphere analysis of microglia, IF, and IHC. Right hemispheres were snap-frozen right after harvesting and stored at -80 °C (Fig 1A).

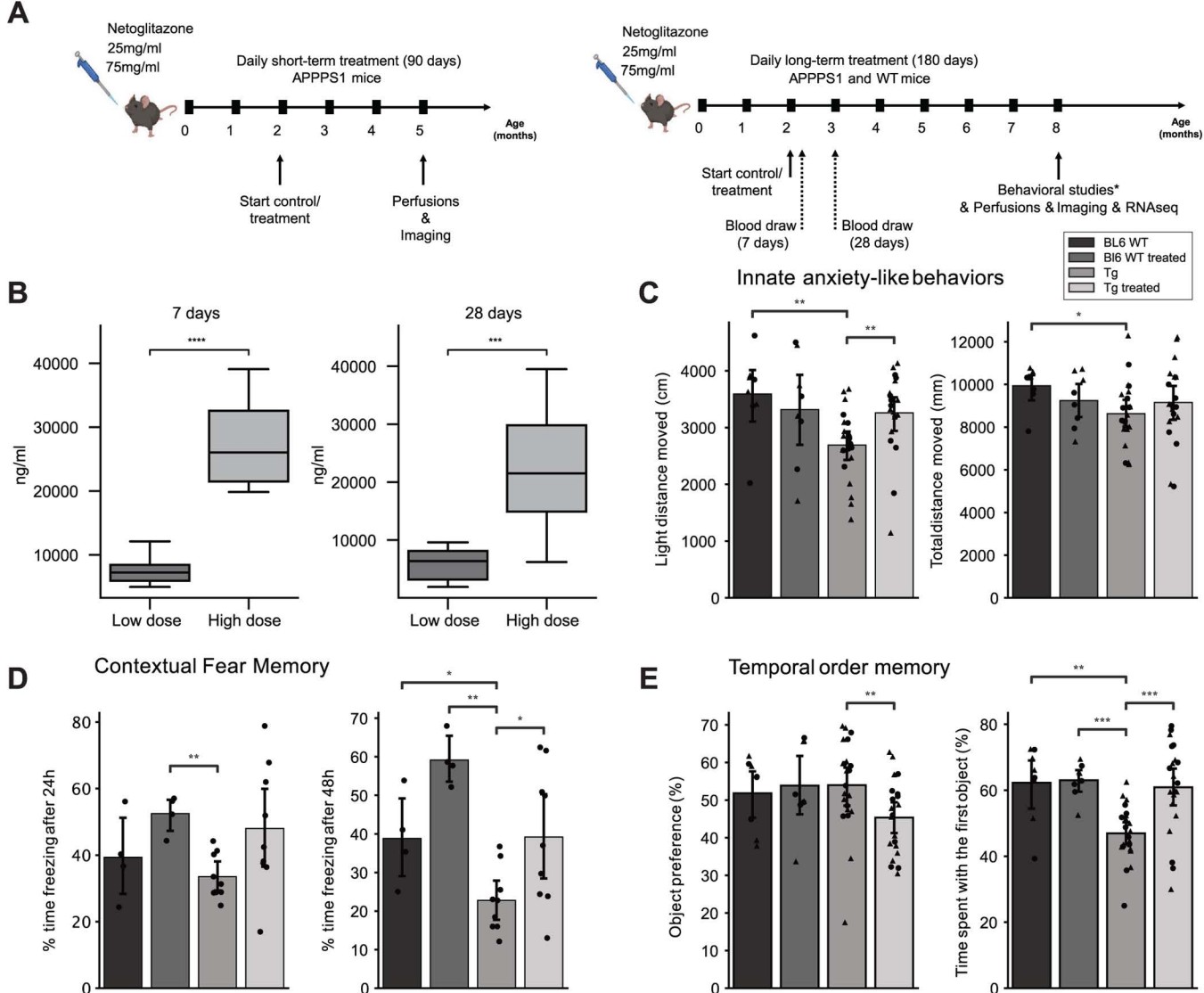

**Fig 1. 180 days of treatment with netoglitazone induces anxiolytic-like effects, alters fear memory and restores deficit in temporal order memory in AD mice.** (A) APPPS1 mice were treated daily with either a high dose (75 mg/ml) or a low dose (15 mg/ml) of netoglitazone or appropriate control (PBS). Treatment started at 2 months of age and continued for either three or six months. Mice were perfused and brains were further analyzed at either five or eight months of age. Behavioral studies and RNAseq analysis were performed only on mice treaded for 180 days with a high dose of netoglitazone. (B) Blood was withdrawn from mice of the long-term treatment cohort seven and twenty-eight days after the start of the treatment. Drug concentration was measured in the plasma. Plasma drug concentration increased in line with time in both dosing cohorts. (C) Behavioral tests: Drug treatment induced anxiolytic-like effects in Tg animals, as indexed by an increased distance moved in the light compartment when compared to untreated Tg animals, as well as to wild type (WT) animals, without affecting general maze exploration. (D) Contextual fear memory: In male mice, long-term drug treatment enhanced expression of contextual fear independent of the genotype after 24h. After 48h, transgenic (Tg) mice displayed a reduction in fear memory, with a treatment-dependent increase, independent of genotype. (E) Tg animals displayed a deficit in temporal order memory, which was restored to control levels by the drug treatment. This was not confounded by changes in the preference towards one of the distinct sets of objects. In graphs C, D, and E triangles represent females, while circles represent males.

## Behavioral studies

Groups of 10 APPPS1 [33] mice and wild-type (WT) mice treated for 180 days with high-dose of netoglitazone (10 mice per group) and respective controls (PBS) were tested approximately 3–4 days prior perfusion for the following behavioral paradigms:

**Light/dark box (LDB) test.** The LDB test was used to measure anxiety-like behavior in mice [34]. The LDB consists of four identical two-way shuttle boxes (30 x 30 x 24 cm; Multi Conditioning System, TSE Systems GmbH, Bad-Homburg, Germany). The boxes are each separated by dark plexiglass walls, which are interconnected by an opening (3.5 × 10 cm) in the partition wall, thus allowing the animal to freely traverse from one compartment to the other. This wall divides the compartment into a dark (1 lux) and a brightly illuminated (100 lux) compartment. Mice were individually placed in the center of the dark compartment and were allowed to move freely for 10 min. The distance moved in the light compartment was assessed as an index of innate anxiety in mice [35] (Fig 1C and S6 Files).

**Spatial recognition memory.** Spatial recognition memory is evaluated by a spatial novelty preference task in the Y-maze [36]. The apparatus was made of transparent Plexiglas and consisted of three identical arms (50 × 9 cm; length × width) surrounded by 10-cm high transparent Plexiglas walls. The three arms radiated from a central triangle (8 cm on each side) and were spaced 120° from each other. Access to each arm from the central area can be blocked by a removable opaque barrier wall. The maze was elevated 90 cm above the floor and positioned in a well-lit room enriched with distal spatial cues. For each retention interval to be tested (see below), the experiment was performed in a different room with a distinct set of extra-maze cues surrounding the Y-maze, to avoid confounds by familiar visual cues. A digital camera was mounted above the Y-maze apparatus. Images were captured at a rate of 5 Hz and transmitted to a PC running the EthoVision tracking system (Noldus Information Technology), calculating the time spent and distance moved in the three arms and center zone of the Y-maze. The test of spatial recognition memory in the Y-maze consisted of two phases, called the sample and choice phases. The allocation of arms (start, familiar and novel arm) to a specific spatial location is counterbalanced across the subjects.

- Sample phase: The animals were allowed to explore two arms (referred to as 'start arm' and 'familiar arm'). Access to the remaining arm ('novel arm') was blocked by a barrier wall door. To begin a trial, the animal was introduced at the end of the start arm and allowed to freely explore both the start and the familiar arms for 5 min. The animals were then removed and kept in a holding cage during the specific retention intervals (see below) prior to the choice phase. The barrier door was removed and the floor was cleaned to avoid olfactory cues.

- Choice phase: Following a specific retention interval (see below), the test animal was introduced to the maze again. During the choice phase, the barrier wall was removed so that the animals could freely explore all arms of the maze for 5 min. The subject was then removed from the maze and returned to the home cage. For each trial, the time spent in each of the three arms was recorded. The relative time spent in the novel arm during the choice phase was calculated by the formula ([time spent in the novel arm/[time spent in all arms]) × 100 and used as the index for spatial novelty preference. In addition, total distance moved on the entire maze was recorded and analyzed to assess general locomotor activity. To manipulate the retention demand in the temporal domain, the interval between the two phases (i.e., sample and choice phases) of the Y-maze test was varied. First, a minimal interval of 1 min was used. The interval between the two phases was then increased to 2h (S1A and S1B Figs. and S7 Files).

**Open field exploration test.** The open-field paradigm was used to study basal locomotor activity [37]. The open field exploration test was conducted in four identical square arenas (40 × 40 × 35 cm high) made of opaque acryl glass. They were in a testing room under diffused lighting (approximately 25 lx as measured in the center of the arenas). A digital camera was mounted directly above the four arenas. Images were captured at a rate of 5 Hz and transmitted to a PC running the Ethovision (Noldus, The Netherlands) tracking system. For measuring basal locomotor activity, the animals

were gently placed in the center of the arena and allowed to explore for 10 min. Distance moved in the entire arena was assessed to index locomotor activity (S1C Fig and S8 File).

**Temporal order memory test.** A temporal order memory test was used as a test for prefrontal cortex-dependent short-term memory. The mouse was first subjected to a training trial, where it was placed in an open field (square arena 40×40×35 cm high) with two copies of a novel object and allowed to explore them for 10 min. After the 10 min exploration, the mouse was placed back into a waiting cage. After a delay of 60 min, the mouse received a second training trial identical to the first, except that two copies of a new novel object will be present. Again, after the second training trial the mouse was placed back into the waiting cage. After a further delay of either 2 h or no delay, the mouse received a test trial identical to the training trials, except that one copy of the object from trial 1 (the old familiar object) and one copy of the object from trial 2 (the recent familiar object) were presented. For each animal, a temporal order memory index was calculated by the formula: ([time spent with phase 1 object]/ [time spent with phase 1 object + time spent with phase 2 object]) * 100. The temporal order memory index was used to compare the animals' capacity to discriminate the relative regency of stimuli [38], with values > 50 signifying a capacity to discriminate between the temporally more remote object presented in sample phase 1 and the temporally more recent object presented in sample phase 2. In addition, the relative amount of time exploring the objects in sample phases 1 and 2 of the test were analyzed to measure object exploration per se (Fig 1E and S9 Files).

**Contextual fear conditioning.** Contextual fear conditioning and extinction were conducted using 4 identical multi-conditioning chambers (Multi Conditioning System, TSE Systems, Bad Homburg, Germany), in which the animals were confined to a rectangular enclosure (30 [length] × 30 [width] × 36 [height] cm) made of black acrylic glass. The chambers were equivalently illuminated by a red house light (30 lux) and were equipped with a grid floor made of 29 stainless rods (4 mm in diameter and 10 mm apart; inter-rod center to inter-rod center), through which a scrambled electric shock could be delivered. Each chamber was surrounded by 3 infrared light-beam sensor systems, with sensors spaced 14 mm apart, allowing movement detection in 3 dimensions. The contextual fear conditioning and extinction test followed protocols established before [39] and consistent of 3 phases, which were each separated 24h apart (see below). During all three phases, the red house light was on at all times. Conditioned fear was expressed as freezing behavior, which was quantified automatically by program-guided algorithms as time of immobility. Habituation and conditioning phase: The animals were placed in the designated test chamber and were allowed to freely explore the chamber for 3 min. This served to habituate the animals to the chamber. Conditioning commenced immediately at the end of the habituation period without the animals being removed from the chambers. For conditioning, the animals were exposed to 3 conditioning trials, whereby each conditioning trial began with the delivery of a 1 second foot-shock set at 0.3 mA and was followed by a 90s rest period. The animals were removed from the chambers and were placed back in their home cages immediately after the last trial. Fear expression phase: The fear expression phase took place 24h and 48h after conditioning when the animals were returned to the same chambers in the absence of any discrete stimulus other than the context. To assess conditioned fear towards to the context, percent time freezing was measured for a period of 6 min. The animals were then removed from the boxes and placed back to their home cages (Fig 1C and S10 Files). Due to an established sex difference in freezing responses as the indicator of conditioned fear, with alternative behavioral responses being present primarily in females [40], we herein focused on male animals in the assessment of fear memory, resulting in the use of fewer animals overall.

## Tissue clearing and staining of Aβ plaques with focused electrophoretic tissue

For whole-brain analysis of Aβ plaques, brains were cleared with focused electrophoretic tissue clearing (FEC) [22]. Briefly: Brains were placed in a custom-built chamber in 8% clearing solution (8% w/w sodium dodecyl sulphate in 200 mM boric acid, pH 8.5) and cleared for approximately 16h at 130 mA current-clamped and at a voltage limit of 60V, at 39.5 °C. Transparency was assessed by visual inspection. Immunofluorescence staining of Aβ plaques was performed in accordance with the protocol described in [22]. Briefly: amyloid plaques were stained with a combination of luminescent conjugated polythiophenes (LCPs), heptamer-formyl thiophene acetic acid (hFTAA), and quadro-formyl thiophene acetic acid

(qFTAA). The combination of these dyes was used to discriminate of neuritic plaques at different maturation states [41]. After staining brains were refractive index (RI) -matched to 1.46 with a modified version of the refractive index matching solution [42] by including triethanolamine [22] (S3 Fig).

## Whole-brain imaging of Aβ plaques

Whole brain images were recorded with a custom-made selective plane illumination microscope (www.mesospim.org) [43]. SPIM imaging was done after clearing and refractive index matching as previously described in [22]. Briefly: the laser/filter combinations for mesoSPIM imaging were as follows: for qFTAA at 488 nm excitation, a 498–520 nm band-pass filter (BrightLine 509/22 HC, Semrock/ AHF) was used as the emission filter; for hFTAA at 488 nm excitation, a 565–605 nm bandpass filter (585/40 BrightLine HC, Semrock/ AHF) was used. Transparent whole-brains were imaged at a voxel size of $3.26 \times 3.26 \times 3 \ \mu m^3$ (X × Y × Z). For scanning a whole brain, 16 tiles per channel were imaged (8 tiles per brain hemisphere). After the acquisition of one hemisphere, the sample was rotated and the other hemisphere was then acquired. The entire process was followed by stitching [44] (S3 Fig).

## Tissue clearing and whole-hemisphere staining of microglia with DISCO

Mouse hemispheres were stained for microglia using a modified version of the iDISCO protocol [45]. Deparaffination was performed using a custom-developed protocol as part of the aDISCO protocol (unpublished). Paraffin-embedded mouse hemispheres were melted for 1 hour at 60°C, followed by incubation in xylene for 1 hour at 37°C and 65 rpm and for 1 hour at room temperature (RT) and 40 rpm. Rehydration was performed by serial incubations of 100%, 95%, 90%, 80%, 70%, 50%, and 25% ethanol (EtOH) in ddH$_2$O, followed by incubation in PBS overnight at RT and 40 rpm. Samples were again dehydrated in serial incubations of 20%, 40%, 60%, 80% methanol (MeOH) in ddH$_2$O, followed by 2 times 100% MeOH, each for 1 hour at RT and 40 rpm. Pre-clearing was performed in 33% MeOH in dichloromethane (DCM) overnight at RT and 40 rpm. After washing 2 times in 100% MeOH each for 1 hour at RT and then 4°C at 40 rpm, bleaching was performed in 5% hydrogen peroxide in MeOH for 20 hours at 4°C and 40 rpm. Samples were rehydrated in serial incubations of 80%, 60%, 40%, and 20% MeOH in in ddH$_2$O, followed by PBS, each for 1 hour at RT and 40 rpm. Permeabilization was performed by incubating the mouse hemispheres 2 times in 0.2% TritonX-100 in PBS each for 1 hour at RT and 40 rpm, followed by incubation in 0.2% TritonX-100 + 10% dimethyl sulfoxide (DMSO) + 2.3% glycine + 0.1% sodium azide (NaN3) in PBS for 5 days at 37°C and 65 rpm. Blocking was performed in 0.2% Tween-20 + 0.1% heparin (10 mg/ml) + 5% DMSO + 6% donkey serum in PBS for 2 days at 37°C and 65 rpm. Samples were stained gradually with primary polyclonal rabbit-anti-Iba1 antibody (Wako, 019–19741) 1:400, followed by secondary polyclonal 647-conjugated donkey-anti-rabbit antibody (ThermoFisher, A-31573) in 0.2% Tween-20 + 0.1% heparin + 5% DMSO + 0.1% NaN3 in PBS (staining buffer) in a total volume of 1.5 ml per sample every week for 2 weeks at 37°C and 65 rpm. Washing steps were performed in staining buffer 5 times each for 1 hour, and then for 1–2 days at RT and 40 rpm. Clearing was started by dehydrating the samples in serial MeOH incubations as described above. Delipidation was performed in 33% MeOH in DCM overnight at RT and 40 rpm, followed by 2 times 100% DCM each for 20 minutes at RT and 40 rpm. Refractive index (RI) matching was achieved in dibenzyl ether (DBE, RI = 1.56) for 4 hours at RT. 3D stacks of cleared mouse hemispheres were acquired using the mesoSPIM light-sheet microscope [43] (www.mesospim.org) at 2X zoom with a field of view of 1.3 cm and isotropic resolution of 3 μm/voxel. To image the microglia a 640 nm laser and a F76 647SG long pass filter were used. Imaged tiles were stitched together [44] and raw data were post-processed using Fiji (Image J, 1.8.0_172 64 bit) and Imaris (Oxford Instruments, 9.8.0) (S3 Fig).

## 2D immunofluorescence staining of Aβ plaques with antibody

Slices from formalin fixed and paraffin embedded brain tissue from 180 day-treated APPPS1 mice (n = 3) were stained for Aβ plaques. Slices were stained with mouse anti-human Aβ$_{1-16}$ antibody (6E10, Biolegend SIG-39320, 1:200) after antigen retrieval with 10% formic acid. Slices were blocked with M.O.M. Kit (BMK-2202) and the primary antibody was detected

with Alexa-488 conjugated goat anti-mouse IgG (Invitrogen A-11005, 1:1000 dilution) followed by diamidino-phenylindole (DAPI) staining. Slices were imaged with a Leica SP5 confocal microscope. Nuclei and plaques were imaged with a 10X/0.25 (numerical aperture 0.4). NA dry objective, using the following settings: 405 nm excitation for DAPI (nuclei) and 488nm excitation for amyloid. The dynamic range of images was adjusted consistently across images. Three different cortical regions and one thalamic region were selected per slice (2 slices per sample). Pixels representing the region of interest were classified and counted as plaques (6E10-Alexa488 positive) or background (6E10-Alexa488 negative) with a manually trained (trained on ten images) pixel classifier in ILASTIK [46] and ImageJ. Hypothesis testing was done with a 2-tailed T-test (S4 Fig).

## 2D immunohistochemistry staining of Aβ plaques with antibody

Slices from formalin fixed and paraffin embedded brain tissue from 180 day-treated APPPS1 mice (n = 3) were stained for Aβ plaques. 6-μm-thick paraffin sections (3 sections per mouse) were deparaffinized through a decreasing alcohol series. Slices were stained with mouse anti-human $Aβ_{1-16}$ antibody (6E10, Biolegend SIG-39320, 1:200) and detected using an IVIEW DAB Detection Kit (Ventana). Sections were imaged using a Zeiss Axiophot light microscope. For the quantification of plaque staining in the whole section, pixels were classified and counted as plaques (Aβ positive) or background (Aβ negative) with a manually trained (trained on five images) pixel classifier in ILASTIK [46], and ImageJ. Hypothesis testing was done with a 2-tailed T-test (S4A Fig).

## Drug distribution measurements in plasma

7 days and 28 days post dosing, serial blood samples (~50 μL) were taken from the tail vein of individual animals and delivered into labelled Safe-lock Eppendorf 1.5 mL clear (e.g., T9661 Sigma Aldrich) containing Na-heparin as the anticoagulant (1000iU, 2 μL per vial). The samples was held on wet ice for a maximum of 30 minutes while sampling of all the animals in the cohort was completed. The blood samples were centrifuged for plasma (4°C, 2000-3000g for 8 min) and 25 μL of the resulting plasma were analyzed. Drug concentration in plasma was calculated by Parmidex, London, UK (Fig 1B).

## Computational and statistical analysis for whole-brain Aβ quantification

The following computations were performed using custom scripts written in Python and R [47] as well as existing third-party libraries as previously described [22]. Briefly, the 2-channel (498–520 nm and 565–605 nm) sub-stacks for each brain hemisphere were first stitched together with TeraStitcher [44]. The result was down sampled from the acquired resolution (3.26 μm lateral, 3 μm depth) to an isotropic 25 μm resolution and then registered to the Allen Brain Atlas 25 μm average anatomical template atlas [48]. The 565–605 nm channel at its original resolution was used to determine the locations of aggregates of amyloid-β stained with qFTAA and hFTAA. A random forest classifier was used to classify each voxel as either "belonging to a plaque" or "background" using the open-source Ilastik framework [46] as described in [22]. After down-sampling each aggregate center to 25-μm resolution and applying the optimized registration transformation, the number of aggregates were counted at each voxel in this atlas space and smoothed heatmaps were generated by placing a spherical ROI with 15-voxel diameter (= 375μm) at each voxel and summing the plaque counts within the ROI, as described in [22] (S3A and S3B Figs).

Voxel-level statistics across treated and control brains involved running a two-sided t-test at each heatmap voxel across the two groups. The three-dimensional statistical maps were adjusted using the threshold-free cluster enhancement method [22]. These adjusted p-value maps were then binarized with a threshold of 0.05 for subsequent analysis or visualization. The transformed locations of each plaque were further grouped into 52 different anatomically segmented regions in the Allen Reference Atlas (25) for further statistical analysis between longitudinal groups. These anatomical regions were masked to only include voxels that demonstrated a statistically significant difference (p < 0.05) (Fig 2 and 3).

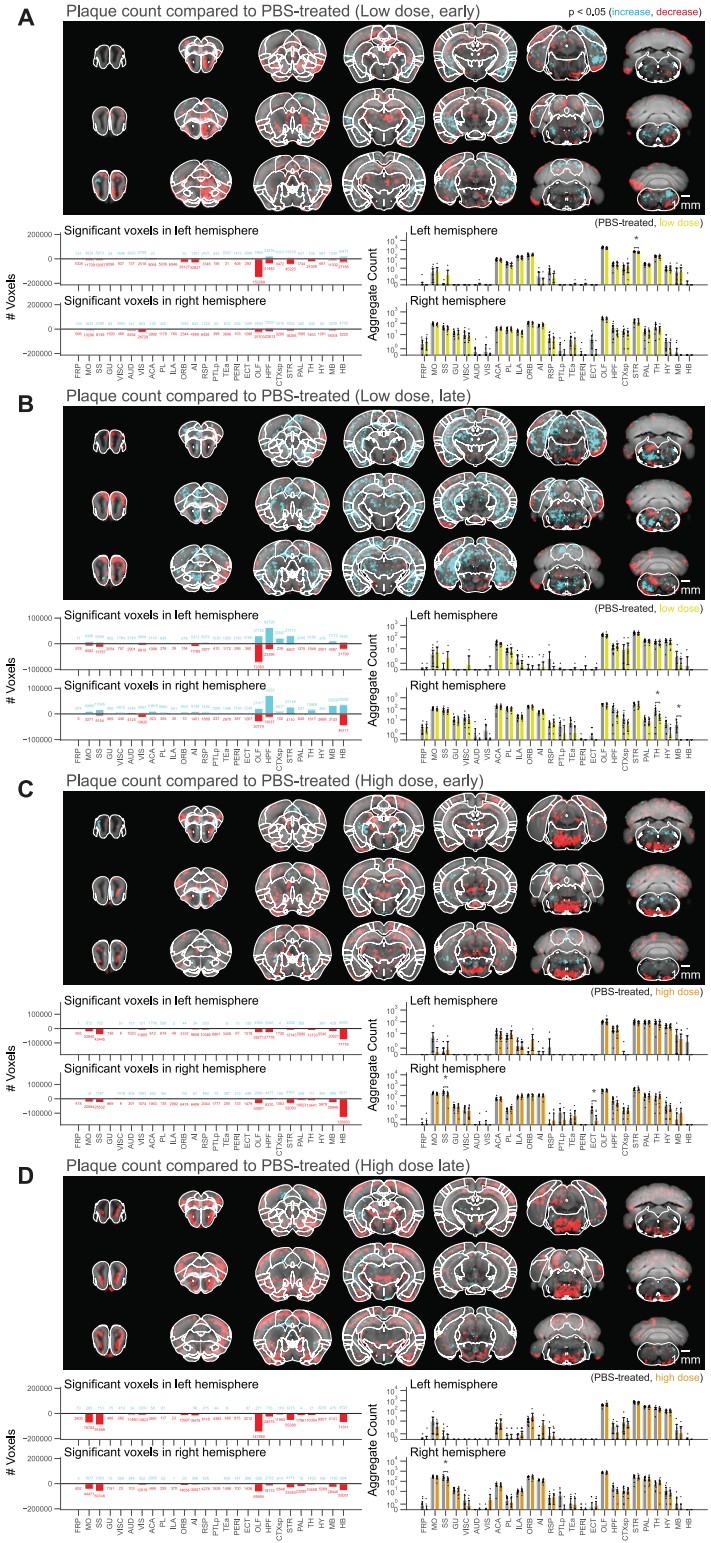

**Fig 2. Voxel-based whole-brain analysis shows regional and dose-dependent effects of netoglitazone in decreasing plaque count.** The figure presents a series of maps and plots illustrating the effects of different treatments on plaque count. Each map represents a 3-dimensional view of statistically affected voxels (p < 0.05), where the red scale indicates a decrease in plaque count, and the cyan scale represents an increase. The reference

atlas is depicted in grey. The maps provide a comprehensive summary of treated and control samples within each cohort, with 6-8 samples per group. Additionally, the plots on the right side display the average plaque count across the cohorts. The figure highlights the regiospecific efficacy unique to each treatment modality: (A) Short-term-treatment with a low dose of netoglitazone shows a patchy effect in reducing plaque count, primarily observed in the olfactory, striatal, and thalamic areas. (B) Long-term-treatment with a low dose of netoglitazone also exhibits a patchy effect in decreasing plaque count. This effect is mainly observed in the olfactory, hindbrain, and visual areas. Notably, there is a patchy increase in plaque count observed in the hippocampal, cortical, striatal, and midbrain areas. (C) Short-term-treatment with a high dose of netoglitazone reveals a significant reduction in plaque count, particularly in the hindbrain, midbrain, striatum, and olfactory areas. (D) Long-term treatment with a high dose of netoglitazone demonstrates a considerable decrease in plaque count, especially in the olfactory, striatum, pallidum, hindbrain, and midbrain regions.

## Computational and statistical analysis for whole-hemisphere microglia quantification

The 640 nm channel in its original resolution was used to determine the spatial density of microglia and all the substacks for each brain hemisphere were first stitched together with Terastitcher [44]. Advanced filtering techniques implemented in Python and C were used within a custom pipeline, available on github (https://github.com/aecon/3D-microglia-netoglitazone), aimed at high-speed processing of 3D half-brain mouse datasets. The pipeline consists of three main steps: (i) image restoration, (ii) voxel-based microglia detection, and (iii) regional microglia-density quantification. First, image restoration was performed to alleviate low frequency background (autofluorescence) undulations, and remove high frequency noise at the voxel level introduced during the digitization of the image via the microscope camera [49]. Specifically, background intensity is modelled via Gaussian smoothing of the raw data. The variance is set to 50 voxels, such that it is larger than the typical foreground (microglia) radius, but smaller than the typical radius of background regions with high autofluorescence. The background undulations are removed by dividing the raw data with the smoothed data [50], yielding the normalized data. Lastly, digitization noise is suppressed via a Gaussian smoothing of the normalized image with a small variance of 1 voxel, modelling the intervoxel noise. Microglia detection was performed by applying a minimum threshold on the normalized intensity data. The threshold was set to 1.8 for all samples, chosen such that large and bright microglia are detected, while at the same time noise and regions of high tissue autofluorescence are excluded. This leads to the binarized data where each pixel is classified as background or foreground. Connected foreground voxels are identified as single microglia cells, and thresholds on the minimum possible microglia volume and minimum maximum microglia intensity are applied to eliminate smaller and/or dimmer artifacts. The segmentation results for all 9 samples were validated by a domain expert, through a visual inspection of the detected microglia overlayed on the raw data (S5 Fig).

Spatial distribution of microglia volume was then estimated by mapping the detected microglia on the Allen Reference Atlas. This step was achieved using elastix [51] where the autofluorescence from the plaque channel (565–605 nm) was first down-sampled to the atlas resolution (25 μm per voxel side) and then used to spatially transform the autofluorescence data such that they match the atlas geometry. This process gave an optimized registration transformation per sample. The optimized transformation was then applied on the detected microglia voxels, after down-sampling to the atlas resolution. Assuming that the density is constant over all selected microglia voxels, the total microglia volume per atlas voxel was computed by counting the number of microglia voxels mapped onto each atlas voxel. The quantification of microglia distribution was performed using density plots, depicting the volume of detected microglia inside a cubic pixel with the atlas resolution. Similar to the plaque quantification, smoothed heatmaps were generated by placing a spherical ROI with 15-voxel diameter and taking a weighted sum of the microglia volume within the ROI. Coronal sections of the volume distribution for every sample, were overlayed on the respective slices of the Allen Brain Atlas (Fig 4A). The average microglia volume distribution per group was computed by taking the mean over the samples belonging to each group (Fig 4B). Using the 134 different anatomically segmented regions of the Allen Reference Atlas, the anatomical regions of the detected microglia were identified, and the total volume in six brain regions was computed: brain stem, hippocampus, hypothalamus, cortex, thalamus, cerebellum. The group-wise average microglia volume and corresponding standard deviation per brain region were then computed. To measure the degree of spatial colocalization between microglia cell count change and plaque count change following long-term treatments of either low or high dose drug, thresholded voxel level

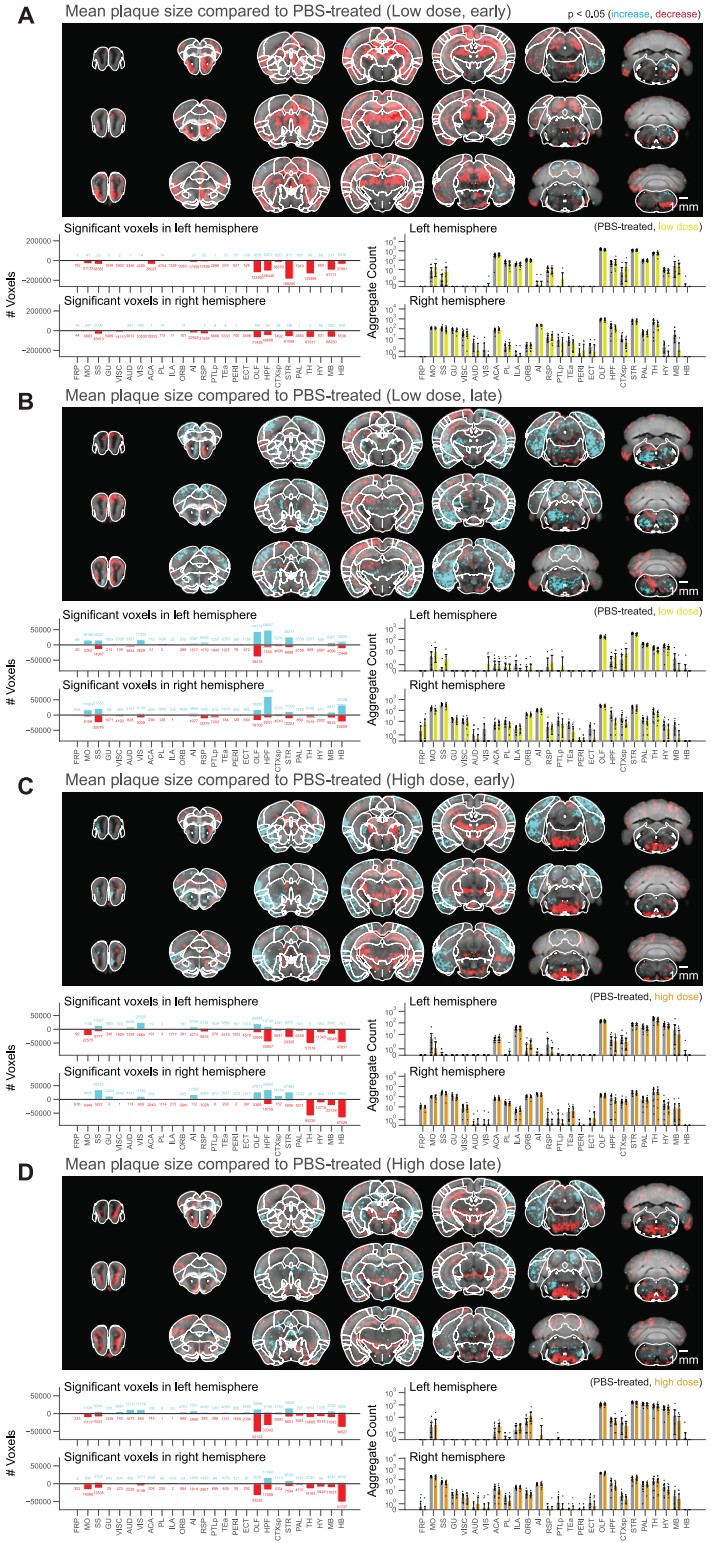

**Fig 3. Voxel-based whole-brain analysis shows regional and dose-dependent effects of netoglitazone in decreasing plaque mean size.** Similar to Figs 2, 3 illustrates a series of maps and plots illustrating the effects of different treatments on plaque size. (A) Short-term-treatment with a low dose of netoglitazone displays a notable patchy effect in reducing plaque size, primarily observed in the olfactory, hippocampal, striatal, thalamic, and midbrain

areas. (B) Long-term-treatment with a low dose of netoglitazone demonstrates a patchy effect primarily in increasing plaque size, mainly observed in the optical cortex, visual area, hippocampus, striatum, and hindbrain. However, there is only a minimal effect on decreasing plaque size. (C) Short-term-treatment with a high dose of netoglitazone reveals a significant reduction in plaque size, especially observed in the hippocampus, striatum, thalamus, hypothalamus, midbrain, and hindbrain. Additionally, there is an increase in plaque size mainly observed in the olfactory, hippocampal, cortical, and striatal areas. (D) Long-term treatment with a high dose of netoglitazone displays a decrease in plaque size, particularly observed in the orbital cortex, olfactory area, hippocampus, thalamus, hypothalamus, midbrain, and hindbrain regions.

statistical maps (p < 0.05, corrected) for each group were first generated in the Allen Coordinate Space. To compare two statistical maps from different groups, we calculate the number of overlapping voxels between the significantly increasing, or decreasing, parts of the first map with the significantly increasing, or decreasing, parts of another map. This results in four distinct comparisons, and a colocalization matrix as depicted in Fig 4B. This analysis was performed for comparing the changes in microglia cell count with plaque changes in the low dose (long-term) treatment group, and separately for comparing the changes in microglia cell count with plaque changes in the high dose (long-term) treatment group.

### RNA sequencing

Group of treated and controls APPPS1 mice (7 high-dose, 3 low-dose and 7 PBS-treated mice) were analyzed for transcriptomic changes. Snap-frozen hemispheres were sectioned in slices of 10 µm, and total RNA was extracted by following a standard RNA extraction protocol (TRIzol Reagent Ref. 15596026). The RNA quality and quantity were assessed using a spectrophotometer, and only high-quality RNA samples were used for subsequent RNA-seq library preparation. The RNA-seq libraries were prepared using a library preparation kit compatible with the sequencing platform (Nova Seq Illumina Library) following the manufacturer's instructions. Subsequently, the libraries were sequenced on a high-throughput sequencing instrument, generating millions of reads per sample.

Post-processed DEGs were visualized with volcano plot showing statistical significance (P-value) versus magnitude of change (fold change). Statistical threshold has been applied prior data visualization (absolute log2 fold change > 0.5 and pvalue < 0.005). Customized script was used to generate the related plot by using R and RStudio platform. For data wrangling, the tidyverse, tidyr and dyplr R packages have been used, while for data visualization ggplot2 and ggpubr packages were used. Differential downregulated genes are shown in green, whereas upregulated genes in magenta (Fig 5).

## Results

### Long-term treatment with netoglitazone significantly reduces cognitive deficits in APP/PS1 mice

Previous findings showed that netoglitazone decreases the aggregation of Aβ fibrils in cell-free in vitro and in vivo models of C. elegans, as well as penetrates the blood-brain barrier of mice when administered orally [32]. Thus, we investigated the removal of neuritic plaques by netoglitazone in APPPS1 mice [33]. Following oral administration, plasma drug levels reached 6297 ng/ml and 6272 ng/ml (SD = 6736) at 7 and 28 days, respectively, for the low-dose treatment. In the high-dose treatment, plasma concentrations measured 21834 ng/ml and 13544 ng/ml (SD = 2364) at the same time points. These findings indicate that the drug attains stable plasma levels over the course of a month (Fig 1B). We then investigated whether high doses of netoglitazone could improve behavioral outcomes in APPPS1 mice [33]. The results showed that netoglitazone treatment led to significant improvements in contextual fear memory, innate anxiety-like behaviors, and temporal order memory, compared to non-treated mice (Figs 1 C, D, and E). However, no improvements were observed in basal locomotor activity or spatial recognition memory (S1 Fig).

In the light/dark box paradigm treated APPPS1 mice showed an increased distance moved in the light compartment as compared to untreated animals. These results indicate that netoglitazone has anxiolytic effects in APPPS1 animals while not affecting their general maze exploration (Fig 1C).

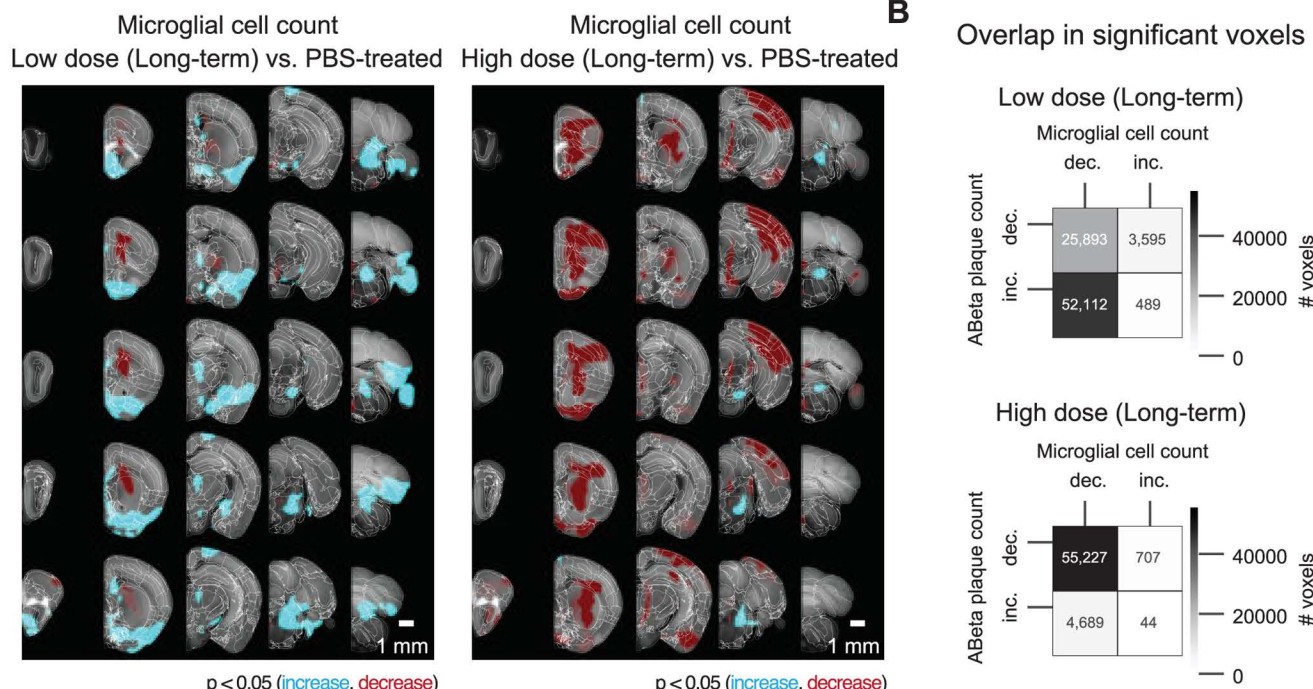

**Fig 4. Voxel-based whole-hemisphere analysis shows regional and dose-dependent decrease of microglia.** (A) Each 3-dimensional map of statistically affected voxels ($p < 0.05$, with the red scale, representing the significance in decrease of microglia, and the cyan scale, representing the increase in microglia; reference atlas is grey) summarizes all the treated and control samples within a cohort (3 samples per group). These maps illustrate that the effectiveness of reducing microglia is specific to particular regions and varies based on the dosage of the treatment. When netoglitazone is administered in low doses over long-term treatment, it only has a limited impact on reducing the overall volume of microglia. However, when high doses of netoglitazone are administered for long-term, it produces a significant and scattered effect, leading to a noticeable reduction in microglia volume in cortical and hippocampal regions. (B) Colocalizing a microglia statistical map with an Aβ plaque statistical maps allows for calculations of the number of overlapping voxels with statistically significant increase or decrease across the two maps. For the late low dose treatment, an increase in Aβ plaque count was colocalized with decreased microglia. For the late high dose treatment, the decrease in Aβ plaque count was also colocalized with decreased microglia.

In the contextual fear memory test, results showed that after 24 hours, netoglitazone-treated mice displayed significantly more freezing behavior compared to the PBS-treated mice, regardless of genotype. 48 hours after conditioning, both APPPS1 and WT mice treated with netoglitazone exhibited increased freezing time compared to their respective PBS-treated controls, indicating a potential deficit in fear memory in untreated animals (Fig 1D).

In the temporal order memory test the results showed that APPPS1 mice exhibited impaired temporal order memory, whereas no such impairment was observed in WT mice. However, the impairment in temporal order memory observed in the netoglitazone-treated APPPS1 group was restored to control levels (Fig 1E).

These findings provide evidence for the efficacy of netoglitazone in improving the temporal short-term memory of APPPS1 mice.

Furthermore, we assessed the spatial memory abilities of the animals and we observed no significant differences in the time spent in the novel arm among the different treatments and genotypes, indicating that netoglitazone treatment did not improve short-term spatial memory (S1A and S1B Figs). We further evaluated the effect of netoglitazone on basal locomotor activity. Our findings indicate that netoglitazone treatment led to a trend towards a decrease in basal locomotor activity in WT mice, as evidenced by a reduction in the distance travelled compared to PBS-treated WT mice. However, there was

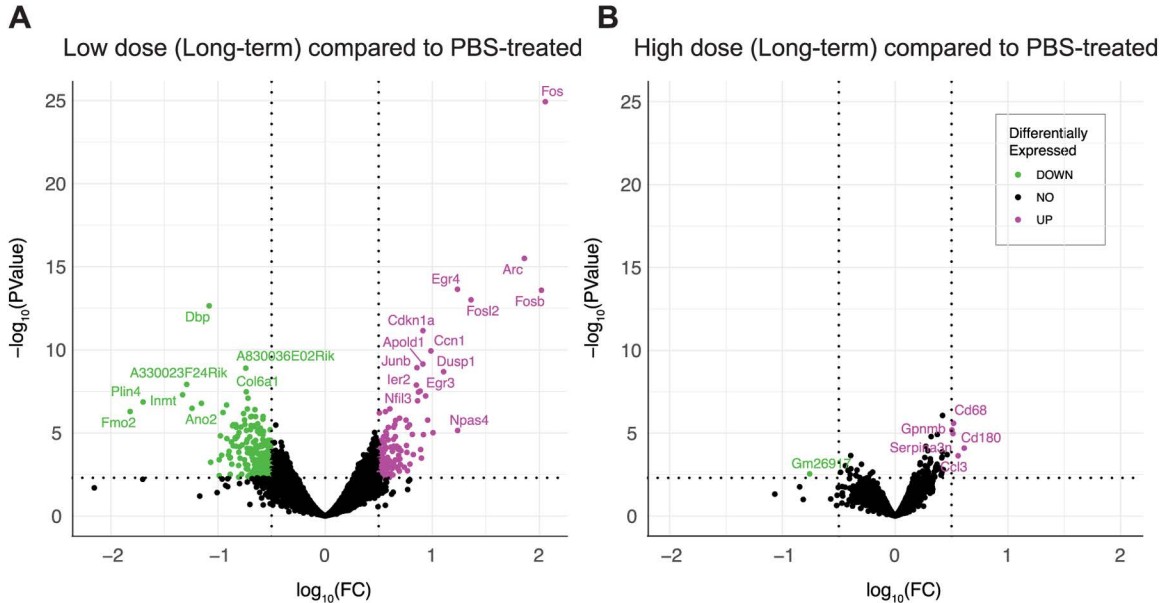

**Fig 5. Gene expression changes are revealed by RNAseq upon long-term treatment with netoglitazone.** Volcano plots depicting the gene regulation effects of long-term treatment with netoglitazone at different doses. The plots include genes that exhibit significant regulation, determined using thresholds of abs|$\log_2$FC|>0.5 and p-value<0.005. (A) The volcano plot for long-term treatment with a low dose of netoglitazone illustrates the significantly regulated genes. Upregulated genes are represented in magenta, while downregulated genes are shown in green. In comparison to the PBS group, animals treated with a low dose of netoglitazone display a notable number of both downregulated and upregulated differentially expressed genes (DEGs). (B) The volcano plot for long-term treatment with a high dose of netoglitazone depicts the significantly regulated genes. Similar to the previous plot, upregulated genes are indicated in magenta, while downregulated genes are displayed in green. However, in contrast to the low dose treatment, animals treated with a high dose of netoglitazone exhibit a minimal number of both downregulated and upregulated DEGs compared to the PBS group.

no effect on basal locomotor activity in APPPS1 mice, as both treated and control mice showed a similar total distance moved (S1C Fig).

**Voxel-based statistics of whole-brain drug efficacy shows regional and dose-dependent effectiveness of netoglitazone in decreasing Aβ aggregates**

To assess the effects of netoglitazone on the amyloid burden in the brain, [22,43,52]. we automatically segmented and regionally quantified plaques using previously described methods[22]. The average plaque count and plaque size within 25-μm voxels were determined for each treatment cohort and presented in a heatmap to show the difference in number and size of plaques between multiple brain areas and treatment conditions (S3A and S3B Figs). To intuitively visualize the therapeutic effect of the two dosages of netoglitazone in the two cohorts, we compared 25-μm voxels corresponding to the average number or plaque size between cohorts (short-term or long-term treatment) and dosages by inferential statistics [22] and depicted them for the whole brain. Voxels revealing a decrease (result of the reduction in the number of accumulated plaques, referred to as "plaque decrease" from here on) or increase (result of the augmentation in the number of accumulated plaques, referred to as "plaque increase" from here on) in plaque count and size were highlighted with two different scale bars. This allowed us to identify voxels significantly altered by the action of netoglitazone and to generate digitally resliced p-value heatmaps (p<0.05) in coronal sections. (Fig 2, Fig 3, and S Videos). We also investigated the differences in plaque count and mean size between treatment groups in 52 neuroanatomical regions defined by the Allen Brain Atlas [53] (Fig 2, Fig 3, and S1 File). In the low-dose cohorts, heatmaps of p-values at voxel level showed that the effect of short-term treatment with netoglitazone promoted a decrease in the count of plaques in

certain areas belonging primarily to the olfactory, striatal, and thalamic areas compared to PBS-treated mice (Fig 2A), while long-term treatment reduced the count of plaques in areas belonging primarily to the olfactory, hindbrain, and visual areas (Fig 2B). However, the low dose long-term treatment also induced a scattered increase in the hippocampal, cortical, striatal, and midbrain areas.

With regard to the high-dose short-term treatment cohort, the drug's effect in reducing the count of plaques was observed extensively in the hindbrain, midbrain, striatum, and olfactory areas (Fig 2C). In a similar pattern, a decrease in the count of plaques following long-term high-dose treatment was observed in certain areas of the olfactory, striatum, pallidum, hindbrain, and midbrain regions (Fig 2D). In addition, we studied the effect of the drug on mean plaque size. Using voxel-p-value maps, we observed that a short-term treatment with low-dose netoglitazone resulted in a strong and widespread decrease in plaque size in olfactory, hippocampal, striatal, thalamic, and midbrain areas (Fig 3A). When the low dose of drug was administered for long-term, an increase in plaque size was mainly detected in optical cortex, visual area, hippocampus, striatum, and hindbrain. (Fig 3B). When we administered a high dose of netoglitazone for short-term, we observed a significant decrease in plaque size in hippocampus, striatum, thalamus, hypothalamus, midbrain, and hindbrain, but we also observed a diffuse increase in olfactory, hippocampal, cortical, and striatal areas (Fig 3C). Examining long-term high-dose treatment, we observed a significant decrease in plaque size in orbital cortex, olfactory area, hippocampus, thalamus, hypothalamus, midbrain, and hindbrain regions (Fig 3D). In all the cases described above, we have noticed a parallel distribution of effects in the two hemispheres.

In summary, when given at a high dose over extended periods, netoglitazone was highly effective in reducing both the number and average size of plaques. Conversely, the low dose of netoglitazone showed greater efficacy in reducing plaque count and size when administered for shorter durations. Nevertheless, there were instances where plaque average size increased unfavorably (S3C Fig).

We further validated our 3D histology results with 2D immunohistochemistry (IHC) and immunofluorescence microscopy (IF) (S4A Fig). Whole sagittal slices (for IHC) or smaller selected areas of cortex, hippocampus and thalamus (for IF) were analyzed, and the number of pixels covered by plaques were counted with ILASTIK [46] (S4B Fig) to evaluate the efficacy of the drug in reducing plaque count using quantitative 2D analysis. No significant differences were observed between the treated and control groups. Consequently, we were unable to demonstrate any substantial effect of the drug in reducing the number of plaques in the brain tissue using 2D histology.

## Netoglitazone decreases microglia activation in a dose-dependent manner

To test the effect of netoglitazone on neuroinflammation and gliosis, we [45,48]performed automated microglia segmentation and regional density quantification using a customized computational pipeline aimed at high-speed processing of half-brain mouse datasets. The pipeline consists of three main steps: (i) image restoration aimed at reducing intensity undulations of the background and increasing signal to noise ratio, (ii) microglia segmentation in 3D using intensity-based pixel classification, and (iii) regional microglia-density quantification. More details for the pipeline can be found in the Methods section and the corresponding github repository. We measured the total volume covered by microglia in each hemisphere of the three different cohorts, and observed a significant reduction in Iba1$^+$ microglia in mice treated with the high dose of the drug ($1.98\,mm^3$) compared to those treated with PBS ($4.15\,mm^3$). Surprisingly, mice treated with the low dose of the drug showed an increase in Iba1$^+$ microglia in the brain stem, hypothalamus and thalamus regions compared to PBS (Fig 4A). Next, we analyzed the correlation between the decrease and increase in plaque density and microglia volume in mice that received long-term treatment. On this end, we performed a spatial colocalization of the statistically significant voxel maps for microglia and plaques, and computed the number of voxels that displayed a statistically significant effect (increase or decrease) in both microglia and plaque maps. There was a high correlation between the decrease of plaques and decrease of microglia in mice treated with a high drug dose, while in mice treated with a low dose the decrease in microglia was highly correlated with the increase in plaques (Fig 4B).

## Netoglitazone alters gene expression in a dose-dependent manner

To investigate the gene expression changes promoted by the different netoglitazone treatments, we verified the genetic changes following the two different dosages of netoglitazone after long-term treatment. Treatment with low-dose netoglitazone promoted a significant change in expression of 361 DEGs compared to the PBS-treated mice. 135 genes were upregulated, whereas 226 genes were downregulated (S2-5 Files). Among the top 20 most significantly upregulated DEGs, we found several members of the immediate early gene (IEG) family, including Fos, Arc, Erg4, Fosb, Fosl2, Apold1, Junb, Dusp1, Ier2, Egr3, Nptx2 and Btg2 (Fig 5 and S3 and S5 Files). By contrast, the most significantly downregulated genes seemed to be involved in several independent processes, with the most relevant in regulation of circadian rhythm (DBP), collagen production (Col6a1), modulation of neuronal toxicity (Wsb1), and drug metabolism (Fmo2). The number of DEGs whose expression was significantly altered following high-dose netoglitazone administration, compared with control, was very low. The significantly increased genes turned out to be only five (CD68, Gpnmb, Serpina3n, Cd180, Ccl3), mainly related to inflammatory mechanisms and immune response, while the decreased gene was only one (Gm26917). (Fig 5and S2 and S4 Files).

## Discussion

There is increasing evidence that complete, rapid amyloid clearance could be key to attenuating the progression of AD [54]. Therefore, identifying drugs that effectively disrupt Aβ aggregation could be a valuable strategy to combat AD. PPARγ receptor activation can counteract the pro-inflammatory and pro-oxidant environment in the CNS, central to AD pathogenesis, making them an attractive pharmacological target [55–61]. Pre-clinical and clinical studies have shown that TZDs, a group of PPARγ agonists, can reduce Aβ generation and release, improve learning and memory, and decrease amyloid pathology in a time- and dose-dependent manner [62–66]. However, the therapeutic efficacy of these molecules in clinical trials was found to be modest, possibly due to imprecise assessments of their impact on Aβ plaque load [67–71]. More comprehensive methods may be required to evaluate the effectiveness of these drugs.

Here, we aimed to evaluate the efficacy of the experimental anti-diabetic drug netoglitazone, a PPARγ agonist belonging to the TZD group, in treating, preventing or inhibiting the formation, deposition, accumulation or persistence of amyloid aggregates in vivo [72]. To overcome the limitations of previous studies, we used three-dimensional histology and computational methods to holistically evaluate the efficacy of netoglitazone and to uncover differences at the regional anatomical level. Netoglitazone has shown promise in previous drug-repurposing strategies and in vitro assays by decreasing fibril mass concentration in a dose-dependent manner and improving the fitness of AD worm models (C. elegans) by reducing the number of aggregates that are formed [73]. These results suggest that netoglitazone may have anti-amyloid properties and be effective in treating AD, which motivated us to further investigate its efficacy in vivo.

We studied the effects of netoglitazone in an animal model of AD and found that a high dose can improve fear and temporal memory in APPPS1 mice. The improvement in contextual fear memory suggests that netoglitazone enhances hippocampal-dependent learning, while the restoration of temporal order memory indicates a positive impact on cognitive deficits observed in APPPS1 mice. Interestingly, we also observed increased freezing behavior in WT animals, suggesting that netoglitazone's effects may extend beyond amyloid pathology to broader fear-related memory processes. These effects could be mediated through synaptic plasticity, stress modulation, or other regulatory pathways. Additionally, netoglitazone significantly influenced innate anxiety-like behavior, as shown by differences in distance moved in the light zone between Tg and Tg Treated animals. This suggests a potential role in modulating anxiety-like responses, possibly through its effects on neuroinflammation and stress reactivity. Overall, these behavioral findings indicate that netoglitazone can improve memory performance and influence anxiety-like responses in APPPS1 mice. These effects provide important insights into the functional impact of the treatment.

Using Q3D analysis [22], we discovered that the impact of the drug on plaques depended on the dosage and administration period, leading to both decreased and increased plaques in various brain regions. Our study found that higher

doses of the drug had a greater and more optimal impact in reducing plaque number at long-term treatment, while lower doses were better at reducing both plaque number and size in short-term treatment. Additionally, we found that the boundaries of drug action did not always correspond to historically defined neuroanatomical areas, suggesting the existence of hitherto unrecognized local modifiers within the brain of hosts. Through Q3D, we were able to identify both favorable and unfavorable changes in amyloid quantity that could not have been detected with traditional biochemistry or histology techniques.

Our research suggests that the varying effects of netoglitazone, contingent on the dose and duration of administration, may be linked to the distinctive expression levels of PPARγ receptors in the brain, different cell types, and the specific stage of the disease [74]. To gain insights into this phenomenon, we examined pioglitazone, a drug similar to netoglitazone, which has been found to exert control over PPARγ receptor target genes in neural cells in a dose-dependent and cell-specific manner. The study of pioglitazone sheds light on the underlying mechanisms that determine the beneficial or adverse effects of netoglitazone, varying according to the dosage used and the specific cell types involved [74].

An additional factor that may contribute to the varying effects of netoglitazone is the disparity in PPARγ receptor expression between males and females [74]. Our study did not differentiate between male and female groups. Furthermore, the present study is limited by the spatial resolution of our Q3D mesoSPIM equipment which does not allow for discriminating structures smaller than 3 μm isotropic. For this reason, Q3D could be optimally combined with orthogonal techniques such as single-cell sequencing and spatial transcriptomics, thereby providing a comprehensive and precise descriptions of spatial drug responses.

Our 3D histology study revealed that the efficacy of netoglitazone in reducing amyloidosis exhibits spatial-temporal specificity. This intriguing finding suggests that the drug's effectiveness in combating amyloid plaques might vary based on the location within the brain and the stage of disease progression. The implications of these observations extend beyond netoglitazone and may have generalizable implications for anti-amyloid therapies.

Several factors could contribute to these regional differences in treatment efficacy. First, differential drug distribution across the brain is a key factor. The permeability of the blood-brain barrier, regional blood flow, and the presence of specific transporters can all lead to uneven drug concentrations in different brain areas, potentially resulting in region-specific effects [75]. Additionally, amyloid pathology itself does not uniformly affect the brain; certain regions, like the hippocampus and cortex, are more prone to amyloid deposition, which can vary in density and composition across different areas [76]. In the APPPS1-Jucker mouse model, however, the high amyloid plaque burden in these regions is largely attributed to the use of the Thy1-promoter, which drives transgene expression predominantly in neurons located in these areas, leading to region-specific amyloid accumulation [33]. Finally, the inherent asymmetry in brain structure and function could also play a role in AD. The lateralization of certain brain functions and the asymmetric patterns of neurodegeneration and plaque formation, commonly seen in AD, can influence treatment outcomes. This asymmetry may partly explain why netoglitazone exhibits differential efficacy in reducing plaques on one side of the brain compared to the other [77]. Studies have demonstrated asymmetric cortical thinning and amyloid deposition in AD patients, further supporting the idea that neurodegenerative diseases often exhibit significant asymmetry that may influence therapeutic responses [78–81].

While asymmetry is a well-documented feature of AD pathology, it is also observed in other neurodegenerative conditions, such as Parkinson's disease (PD). In PD, asymmetric motor symptoms and neuropathological changes are common and can influence how patients respond to treatment [82–85]. These examples from PD suggest that asymmetry in neurodegeneration, whether motor or cognitive, could be a broader feature of neurodegenerative diseases, underscoring the importance of considering such asymmetries when developing and assessing therapeutic interventions.

In summary, our study highlights the intricate relationship between netoglitazone's effects and dosage, administration duration, cell type, and disease stage. While we observed sex differences in behavioral responses and receptor expression, the impact of sex dimorphism on Aβ plaque load was not explicitly assessed in this study. It remains unclear whether sex-specific differences in Aβ plaque deposition in APPPS1 mice play a significant role in modulating the effects of

therapeutic interventions. Future studies focusing on stratifying data by sex could help elucidate these potential disparities and provide additional insight into how biological sex influences therapeutic outcomes.

Understanding these multifaceted factors can contribute to optimizing therapeutic approaches and uncovering novel treatment strategies for amyloid-related disorders and possibly other neurodegenerative diseases.

Inflammation and gliosis are histological hallmarks of AD and can be observed in APPPS1 mice from an early age on [86]. Aβ plaque-associated reactive microgliosis is seen in rodent models of AD and human cases, indicating that Aβ deposition leads to microglial activation [87–90]. PPARγ agonists have been shown to inhibit microglial activation and inflammation, making them a potential therapeutic option for AD [91–93]. Q3D allows for precisely quantifying the changes in microglia volume following long-term treatment with netoglitazone. We found that high-dose netoglitazone significantly reduced the total volume of microglia throughout the brain, particularly in the cortex, and this correlated with a decrease in Aβ plaques. In contrast, low-dose netoglitazone had a mixed effect on microglia in a spatially-dependent manner. These findings suggest that a long-term high dose of netoglitazone may reduce inflammation and enhance the phagocytic activity of microglia, which facilitates the removal of Aβ deposits.

We measured gene expression changes in APPPS1 mice treated with different doses of netoglitazone and compared them to PBS. Among the genes that showed a significant difference between low-dose netoglitazone and PBS, the 20 most upregulated genes were immediate early genes (IEGs) associated with neuronal plasticity and memory formation [94]. This suggests that low-dose treatment stimulated a stress and inflammation response, potentially due to the drug's localized efficacy throughout the brain [95–99]. On the other hand, high-dose treatment led to a small number of differentially expressed genes, mostly related to microglia activation and immune defense mechanisms, indicating a decrease in amyloidosis and inflammation throughout the brain [100–105]. These findings are consistent with our observations from the whole-brain maps.

The interplay between various brain cells and amyloidosis is complex [106]. For example, we found that Serpina3n, a marker of disease-associated astrocytes [107] and oligodendrocytes [108], was upregulated following high-dose netoglitazone treatment, indicating that these cells may be pharmacodynamically involved [109]. Importantly, our study showed that high-dose netoglitazone treatment is associated with plaque reduction and a decrease in microglia; however, the remaining microglia shifted to a phagocytic CD68-high state [110]. This is consistent with findings indicating that lower microglia counts reduce plaque seeding and that phagocytosis decreases amyloid levels [111,112], potentially highlighting the therapeutic mechanism of netoglitazone.

We also demonstrate that relying on arbitrary or selected 2D sections may lead to ambiguous or incomplete interpretations. While traditional 2D immunohistochemistry has been a standard approach for analyzing amyloid plaques, it provides only a limited, slice-based perspective that may fail to capture the full extent of amyloid deposition and distribution throughout the brain. This approach can overlook intricate spatial patterns of plaque accumulation, resulting in an incomplete understanding of the disease's progression and drug efficacy. In contrast, 3D histochemistry offers a high-resolution, whole-brain view, revealing the total plaque burden, including subtle regional differences and asymmetries. This comprehensive method enables the detection of plaques that could be missed by 2D sectioning. Moreover, 3D histochemistry allows for a more accurate and detailed assessment of treatment effects, such as those seen with netoglitazone, where outcomes might be underestimated or entirely missed using traditional 2D techniques.

Beyond its significance in the evaluation of netoglitazone in AD, the present study showcases Q3D as an advanced technique capable of identifying phenomena that had gone undetected by conventional microscopy.

## Significance statement

Alzheimer's disease (AD) is the most prevalent neurodegenerative disease. Its primary symptom is progressive cognitive decline, which impairs executive brain functions and deprives patients of their autonomy in life. Experimental

and clinical evidence points to the critical pathophysiological role of the amyloid-beta (Aβ) peptide. Despite some limited successes in AD immunotherapy targeting Aβ, AD is still incurable. Here, we use an innovative pipeline for accurate whole-brain measurements of Aβ load to test the efficacy of the antidiabetic compound, netoglitazone. We found that netoglitazone decreases Aβ burden in certain brain areas but not in others. Region-specific assessment of anti-Aβ efficacy may be useful in the development of effective drugs against Alzheimer's disease.

## Supporting information

**S1 Fig. 180 days of treatment with netoglitazone does not improve spatial recognition memory and basal locomotor activity.**
(DOCX)

**S2 Fig. Raw data of Aβ plaques and microglia staining.**
(DOCX)

**S3 Fig. Average number and size of plaque in the different cohorts.**
(DOCX)

**S4 Fig. 2D immunohistochemistry and immunofluorescence of Aβ plaques.**
(DOCX)

**S5 Fig. Segmentation of microglia cells.** Representative images depicting the quality of microglia segmentation.
(DOCX)

**S1 File. Plaque counts.**
(XLSX)

**S2 File. Downregulated genes of high dose treated mice.**
(CSV)

**S3 File. Downregulated genes of low dose treated mice.**
(CSV)

**S4 File. Upregulated genes of high dose treated mice.**
(CSV)

**S5 File. Upregulated genes of low dose treated mice.**
(CSV)

**S6 Files. Behavioral tests.** Lightdark box (LDB) test_LD Box Light Distance Moved and Lightdark box (LDB) test_LD Box Total Distance Moved.
(ZIP)

**S7 Files. Behavioral tests.** Spatial recognition memory_Y Maze 2 hrs delay and Spatial recognition memory_Y Maze No delay.
(ZIP)

**S8 File. Behavioral tests.** Open field exploration test _Open Field Dist Moved.
(ZIP)

**S9 Files. Behavioral tests.** Temporal order memory test _Time Remote and Temporal order memory test _Objects preference.
(ZIP)

**S10 Files. Behavioral tests.** Contextual Fear Conditioning_24h and Contextual Fear Conditioning_48h.
(ZIP)

## Acknowledgments

We thank Wren Therapeutics, Cambridge, UK, for providing netoglitazone.

## Author contributions

**Conceptualization:** Francesca Catto, Daniel Kirschenbaum, Adriano Aguzzi.

**Data curation:** Francesca Catto, Ehsan Dadgar-Kiani, Davide Caredio, Lukas Frick, Ulrike Weber-Stadlbauer.

**Formal analysis:** Ehsan Dadgar-Kiani, Athena E. Economides, Davide Caredio, Lukas Frick.

**Funding acquisition:** Petros Koumoutsakos, Jin Hyung Lee, Adriano Aguzzi.

**Investigation:** Francesca Catto, Daniel Kirschenbaum, Jin Hyung Lee, Adriano Aguzzi.

**Methodology:** Francesca Catto, Ehsan Dadgar-Kiani, Daniel Kirschenbaum, Athena E. Economides, Anna Maria Reuss, Delic Mirzet, Ulrike Weber-Stadlbauer, Sergey Litvinov, Petros Koumoutsakos, Jin Hyung Lee.

**Project administration:** Francesca Catto.

**Resources:** Ulrike Weber-Stadlbauer, Petros Koumoutsakos, Jin Hyung Lee, Adriano Aguzzi.

**Software:** Ehsan Dadgar-Kiani, Athena E. Economides, Sergey Litvinov.

**Supervision:** Petros Koumoutsakos, Jin Hyung Lee, Adriano Aguzzi.

**Validation:** Francesca Catto, Daniel Kirschenbaum, Chiara Trevisan, Adriano Aguzzi.

**Visualization:** Francesca Catto, Ehsan Dadgar-Kiani, Daniel Kirschenbaum, Athena E. Economides, Lukas Frick.

**Writing – original draft:** Francesca Catto, Adriano Aguzzi.

**Writing – review & editing:** Francesca Catto, Ehsan Dadgar-Kiani, Daniel Kirschenbaum, Athena E. Economides, Ulrike Weber-Stadlbauer, Petros Koumoutsakos, Jin Hyung Lee, Adriano Aguzzi.

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
