## [Editor Report · Decision Letter 0]

21 Aug 2024

PONE-D-24-31056

Quantitative 3D histochemistry reveals region-specific amyloid-β reduction by the antidiabetic drug netoglitazone

PLOS ONE

Dear Dr. Catto,

Thank you for submitting your manuscript to PLOS ONE. After careful consideration, we feel that it has merit but does not fully meet PLOS ONE’s publication criteria as it currently stands. Therefore, we invite you to submit a revised version of the manuscript that addresses the points raised during the review process.

We invite you to submit a revised version of the manuscript that addresses the points raised during the initial review by the editor.

Supplementary figures (the paper makes reference to at least 4) appear to be missing.

Reference to figure 5 (lines 731 - 733) appears incorrect.

The number of mice within each group is insufficiently clear, as there appears to be differences in group size depending upon the behavioural outcome measure, e.g., Figure 1C versus Figure 1D. The number of males and females within each group is insufficiently clear.

There appears to be regional sidedness in the pattern to reduced plaques and discussion of this is insufficient.

e.g., the ventrolateral cerebellum left side only (possibly paraflocculus, Fig 2A, images on far right)

e.g., right piriform cortex and right frontal cortex and olfactory tubercle (Figure 2A, images on left, 2nd and 3rd columns).

e.g., right caudal hippocampus (Figure 2 A, middle column, middle right column)

Differences between 3D histochemistry and traditional 2D immunohistochemical results are insufficiently discussed in the body of the manuscript.

If you can satisfactorily address these initial issues, we would be happy to consider an amended version of the paper.

A rebuttal letter that responds to each point raised by the academic editor. You should upload this letter as a separate file labeled 'Response to Initial Review'.A marked-up copy of your manuscript that highlights changes made to the original version. You should upload this as a separate file labeled 'Revised Manuscript with Track Changes'.An unmarked version of your revised paper without tracked changes. You should upload this as a separate file labeled 'Manuscript'.

We look forward to receiving your revised manuscript.

Kind regards,

Miriam Ann Hickey, PhD

Academic Editor

PLOS ONE

Journal Requirements:

4. Please remove your figures from within your manuscript file, leaving only the individual TIFF/EPS image files, uploaded separately. These will be automatically included in the reviewers’ PDF.

6. We are unable to open your Supporting Information file "Wren Behavior Data 291121.pzfx". Please kindly revise as necessary and re-upload.

---

## [Author Response · Author response to Decision Letter 1]

25 Sep 2024

Response to Initial Review Manuscript ID: PONE-D-24-31056 Title: Quantitative 3D histochemistry reveals region-specific amyloid-β reduction by the antidiabetic drug netoglitazone

Dear Dr. Miriam Ann Hickey,

We would like to thank you for your thoughtful evaluation of our manuscript. We appreciate the opportunity to revise and improve our work. Below, we provide detailed responses to each of the points raised by the academic editor and reviewers. We have made the necessary revisions to the manuscript accordingly.

Addressed Comments:

1. Supplementary figures (the paper makes reference to at least 4) appear to be missing.

We sincerely apologize for the oversight in the initial submission. It appears that the figures file was mistakenly uploaded instead of the supplementary figures file. We have now corrected this mistake and uploaded the appropriate supplementary figures in the revised submission (S1-S5 Figs.).

2. Reference to figure 5 (lines 731 - 733) appears incorrect.

Thank you for pointing out this error. We have reviewed the reference and corrected it in the revised manuscript. The reference now correctly points to Figure 4, aligning with the relevant content in the text. The references to the figures are now located on lines 481 and 483.

3. The number of mice within each group is insufficiently clear, as there appears to be differences in group size depending upon the behavioural outcome measure, e.g., Figure 1C versus Figure 1D. The number of males and females within each group is insufficiently clear.

We appreciate the reviewer’s observation regarding the clarity of the number of mice used in each group. To address this concern, we have created a detailed table (Table 1) that provides a clear breakdown of the number of mice used for each figure, including the specific group sizes and the distribution of males and females. The table has been added after the paragraph that cites it on line 123 in the 'Materials and Methods – APPPS1 Mice' chapter of the revised manuscript to ensure full transparency and clarity.

Due to an established sex difference in freezing responses as the indicator of conditioned fear (Figure 1D), with alternative behavioral responses being present primarily in females (Gruene TM, Flick K, Stefano A, Shea SD, Shansky RM. 2015. Sexually divergent expression of active and passive conditioned fear responses in rats. Elife 4: e11352. 10.7554/eLife.1135), we herein focused on male animals in the assessment of fear memory, resulting in the use of fewer animals overall. We have added a few lines of clarification starting from line 269.

4. There appears to be regional sidedness in the pattern to reduced plaques, and discussion of this is insufficient.

o e.g., the ventrolateral cerebellum left side only (possibly paraflocculus, Fig 2A, images on far right).

o e.g., right piriform cortex and right frontal cortex and olfactory tubercle (Figure 2A, images on left, 2nd and 3rd columns).

o e.g., right caudal hippocampus (Figure 2A, middle column, middle right column).

We appreciate the reviewer’s insightful comments on the regional asymmetry in plaque reduction. In response, we have expanded our discussion in the revised manuscript to explore these regional differences in greater detail. We now include a dedicated section, starting from line 788 that examines potential reasons for this asymmetry, such as varying drug distribution, regional differences in amyloid pathology, and the natural asymmetry in brain structure and function. These factors may impact the region-specific effectiveness of netoglitazone.

5. Differences between 3D histochemistry and traditional 2D immunohistochemical results are insufficiently discussed in the body of the manuscript.

We appreciate the reviewer’s comment highlighting the need for a more detailed discussion on the differences between 3D histochemistry and traditional 2D immunohistochemistry. In response, we have expanded the discussion section of the manuscript to address this comparison more thoroughly.

Specifically, we have added a paragraph that directly contrasts the capabilities of these two methodologies. While traditional 2D immunohistochemistry offers valuable insights, it is inherently limited by its slice-based perspective, which can result in sampling biases and an incomplete understanding of plaque distribution across the brain. In contrast, 3D histochemistry allows for whole-brain imaging, providing a more comprehensive view of plaque burden and enabling the detection of regional differences and asymmetries that might be overlooked in 2D analyses. This addition is located in the discussion, immediately after the paragraph where we discuss the advantages of Q3D in detecting both favorable and unfavorable changes in amyloid quantity, starting from line 836, thereby enhancing the overall discussion on the methodologies used in our study. Additionally, we have included a supplementary figure (S4 Fig) in the revised submission, which was previously omitted due to an upload error. This figure may provide further clarity in understanding these distinctions. We apologize for the oversight.

Conclusion:

We have addressed all of the points raised during the review process and believe that these revisions have strengthened our manuscript. We hope that the revised manuscript meets the publication criteria of PLOS ONE and look forward to your feedback.

Sincerely,

Dr. Francesca Catto

Wagistrasse 18, 8952, Schlieren

Zurich, Switzerland

catto@imai.ch

www.imai-medtech.com/

---

## [Editor Report · Decision Letter 1]

27 Sep 2024

PONE-D-24-31056R1Quantitative 3D histochemistry reveals region-specific amyloid-β reduction by the antidiabetic drug netoglitazonePLOS ONE

Dear Dr. Catto,

Thank you for re-submitting your manuscript to PLOS ONE.

The authors were invited to submit a revised version of the manuscript that addressed the points raised during the initial review by the editor. The authors are thanked for responding to the initial comments of the editor.

However, within the body of the re-submitted manuscript, there are numerous examples of this text:

"Error! Reference source not found"

We invite you to submit a revised version of the manuscript, which we will then consider for peer review.

A rebuttal letter that responds to each point raised by the academic editor. You should upload this letter as a separate file labeled 'Response to Reviewers'.A marked-up copy of your manuscript that highlights changes made to the original version. You should upload this as a separate file labeled 'Revised Manuscript with Track Changes'.An unmarked version of your revised paper without tracked changes. You should upload this as a separate file labeled 'Manuscript'.

We look forward to receiving your revised manuscript.

Kind regards,

Miriam Ann Hickey, PhD

Academic Editor

PLOS ONE
---

## [Author Response · Author response to Decision Letter 2]

30 Oct 2024

Response to Initial Review Manuscript ID: PONE-D-24-31056 Title: Quantitative 3D histochemistry reveals region-specific amyloid-β reduction by the antidiabetic drug netoglitazone

Dear Dr. Miriam Ann Hickey,

We would like to thank you for your thoughtful evaluation of our manuscript. We appreciate the opportunity to revise and improve our work. Below, we provide detailed responses to each of the points raised by the academic editor and reviewers. We have made the necessary revisions to the manuscript accordingly.

Addressed Comments:

1. Supplementary figures (the paper makes reference to at least 4) appear to be missing.

We sincerely apologize for the oversight in the initial submission. It appears that the figures file was mistakenly uploaded instead of the supplementary figures file. We have now corrected this mistake and uploaded the appropriate supplementary figures in the revised submission (S1-S5 Figs.).

2. Reference to figure 5 (lines 731 - 733) appears incorrect.

Thank you for pointing out this error. We have reviewed the reference and corrected it in the revised manuscript. The reference now correctly points to Figure 4, aligning with the relevant content in the text. The references to the figures are now located on lines 481 and 483.

3. The number of mice within each group is insufficiently clear, as there appears to be differences in group size depending upon the behavioural outcome measure, e.g., Figure 1C versus Figure 1D. The number of males and females within each group is insufficiently clear.

We appreciate the reviewer’s observation regarding the clarity of the number of mice used in each group. To address this concern, we have created a detailed table (Table 1) that provides a clear breakdown of the number of mice used for each figure, including the specific group sizes and the distribution of males and females. The table has been added after the paragraph that cites it on line 123 in the 'Materials and Methods – APPPS1 Mice' chapter of the revised manuscript to ensure full transparency and clarity.

Due to an established sex difference in freezing responses as the indicator of conditioned fear (Figure 1D), with alternative behavioral responses being present primarily in females (Gruene TM, Flick K, Stefano A, Shea SD, Shansky RM. 2015. Sexually divergent expression of active and passive conditioned fear responses in rats. Elife 4: e11352. 10.7554/eLife.1135), we herein focused on male animals in the assessment of fear memory, resulting in the use of fewer animals overall. We have added a few lines of clarification starting from line 269.

4. There appears to be regional sidedness in the pattern to reduced plaques, and discussion of this is insufficient.

o e.g., the ventrolateral cerebellum left side only (possibly paraflocculus, Fig 2A, images on far right).

o e.g., right piriform cortex and right frontal cortex and olfactory tubercle (Figure 2A, images on left, 2nd and 3rd columns).

o e.g., right caudal hippocampus (Figure 2A, middle column, middle right column).

We appreciate the reviewer’s insightful comments on the regional asymmetry in plaque reduction. In response, we have expanded our discussion in the revised manuscript to explore these regional differences in greater detail. We now include a dedicated section, starting from line 788 that examines potential reasons for this asymmetry, such as varying drug distribution, regional differences in amyloid pathology, and the natural asymmetry in brain structure and function. These factors may impact the region-specific effectiveness of netoglitazone.

5. Differences between 3D histochemistry and traditional 2D immunohistochemical results are insufficiently discussed in the body of the manuscript.

We appreciate the reviewer’s comment highlighting the need for a more detailed discussion on the differences between 3D histochemistry and traditional 2D immunohistochemistry. In response, we have expanded the discussion section of the manuscript to address this comparison more thoroughly.

Specifically, we have added a paragraph that directly contrasts the capabilities of these two methodologies. While traditional 2D immunohistochemistry offers valuable insights, it is inherently limited by its slice-based perspective, which can result in sampling biases and an incomplete understanding of plaque distribution across the brain. In contrast, 3D histochemistry allows for whole-brain imaging, providing a more comprehensive view of plaque burden and enabling the detection of regional differences and asymmetries that might be overlooked in 2D analyses. This addition is located in the discussion, immediately after the paragraph where we discuss the advantages of Q3D in detecting both favorable and unfavorable changes in amyloid quantity, starting from line 836, thereby enhancing the overall discussion on the methodologies used in our study. Additionally, we have included a supplementary figure (S4 Fig) in the revised submission, which was previously omitted due to an upload error. This figure may provide further clarity in understanding these distinctions. We apologize for the oversight.

6. Within the body of the re-submitted manuscript, there are numerous examples of this text: "Error! Reference source not found".

Thank you for bringing this issue to our attention. We sincerely apologize for the oversight. We have carefully reviewed the manuscript and corrected all instances of "Error! Reference source not found" by properly adjusting the references throughout the document.

Conclusion:

We have addressed all of the points raised during the review process and believe that these revisions have strengthened our manuscript. We hope that the revised manuscript meets the publication criteria of PLOS ONE and look forward to your feedback.

Sincerely,

Dr. Francesca Catto

Wagistrasse 18, 8952, Schlieren

Zurich, Switzerland

catto@imai.ch

www.imai-medtech.com/

---

## [Decision Letter · Decision Letter 2]

3 Dec 2024

PONE-D-24-31056R2Quantitative 3D histochemistry reveals region-specific amyloid-β reduction by the antidiabetic drug netoglitazonePLOS ONE

Dear Dr. Catto,

Thank you for submitting your manuscript to PLOS ONE. After careful consideration, we feel that it has merit but does not fully meet PLOS ONE’s publication criteria as it currently stands. Therefore, we invite you to submit a revised version of the manuscript that addresses the points raised by the Reviewers during the review process.

We look forward to receiving your revised manuscript.

Kind regards,

Miriam A Hickey, PhD

Academic Editor

PLOS ONE

Additional Editor Comments:

Please respond to all reviewer comments.

Reviewers' comments:

Reviewer's Responses to Questions

**Comments to the Author**

1. If the authors have adequately addressed your comments raised in a previous round of review and you feel that this manuscript is now acceptable for publication, you may indicate that here to bypass the “Comments to the Author” section, enter your conflict of interest statement in the “Confidential to Editor” section, and submit your "Accept" recommendation.

Reviewer #1: (No Response)

Reviewer #2: (No Response)

2. Is the manuscript technically sound, and do the data support the conclusions?

Reviewer #1: Partly

Reviewer #2: Yes

3. Has the statistical analysis been performed appropriately and rigorously?

Reviewer #1: Yes

Reviewer #2: Yes

4. Have the authors made all data underlying the findings in their manuscript fully available?

Reviewer #1: Yes

Reviewer #2: Yes

5. Is the manuscript presented in an intelligible fashion and written in standard English?

Reviewer #1: Yes

Reviewer #2: Yes

6. Review Comments to the Author

Reviewer #1: The authors performed experiments to assess the modulatory effects of netoglitazone on Aβ plaque burden in vivo. The authors proposed that netoglitazone administration reduces Aβ plaque deposition and highlights the strength of volumetric assessment of Aβ plaque pathology in preclinical AD research. While the addition of information regarding asymmetrical plaque deposition is intriguing, there are still some concerns regarding the data presentation of the manuscript in its current form. See comments below:

1. Introduction – Regarding the amyloid cascade hypothesis, could the authors comment on the detrimental role that Aβ oligomers also play in inducing neuronal deficits? Although plaques seemingly exert physical detriments to the parenchyma and cells proximal to them, it remains debated whether an overabundance of oligomeric Aβ not bound in plaques would be more detrimental to neural functions. This could also explain why Aβ plaque deposition does not correlate well with cognitive decline, as it could be possible that plaques act as a reservoir for oligomeric Aβ from disrupting synaptic and other neural functions.

2. Methods:

a. Could the authors provide details on how the APPPS1 mice were bred? For instance, what is the sex of the transgene carrier for all breeding? A consistent reporting of breeding schemes would benefit readers and encourage transparency and reproducibility. Moreover, the usage of the term gender should be replaced with sex.

b. Could the authors please indicate the different sex of animals (different symbols) used in graphs where pooling of male and female animals was performed? The authors noted in this revised manuscript that sex difference in freezing responses occur, but could the authors also comment on whether sex dimorphism in Aβ plaque load is evident between male and female APPPS1 mice? If sex dimorphism is evident, should the assessment of plaque load not be stratified further based on sex? In fact, the authors also openly stated in the discussion section that there is a disparity in PPARγ receptor expression between males and females.

3. Results:

a. Figs 2 and 3 – The significant voxels and average values are very small and rather difficult to read. Is it possible to increase the size of these panels to improve legibility? Moreover, do early and late mean short-term (90 days) and long-term (180 days)? If so, the caption in the figures should be modified accordingly to improve clarity.

b. Figs 2 – Could the authors elaborate on how significant voxels sometimes do not correspond to similar statistical differences between treated and untreated groups in the average count? Should conclusions not be drawn from the average count instead? Lastly, does the average count not correspond with the 2D analysis in Fig S4 which shows unchanged plaque load?

c. Fig 3 – Could the authors comment on why the graphs on the right in this figure depict the average plaque count again as opposed to the average plaque size? Are the average plaque counts per region plotted to again show differences caused by the treatment in a separate cohort of animals? From the table listing the number of experimental animals, it seems that the same mice were used to generate the data from Fig 2 and Fig 3, but the average count values from both figures do not match. Could the authors please clarify?

d. Fig 4A – In terms of data presentation, the white outline of the brain subregions seems to be too thick and hence distracting from the colors annotating the significant increases and decreases. If possible, could the authors please reduce the outline thickness or reduce the number of regions annotated to only show major subregions as in Figs 2 and 3?

e. Fig 5 – In the main text, the authors described an abundance of DEGs associated with the high-dose treatment while DEGs are limited in mice that received low-dose treatment. In the figure, however, this is the opposite. Could the authors please clarify?

4. Discussion:

a. The authors stated that the hippocampus and the cortex are prone to amyloid deposition. However, this statement seems misleading as the high plaque load in these regions can be directly linked to regions of Thy1-promoter activity (PMID: 12112467), which is the promoter used in the APPPS1-Jucker line. Hence, high plaque load in these regions is presumably a direct effect of transgene expression in a subset of neurons in said regions.

Reviewer #2: This study aims to interrogate the effect of netoglitazone in an animal model of Alzheimer's disease. To evaluate its effect, the authors perform behavioral tests and whole-brain immunohistochemistry. The authors found changes in behavior and a dose-dependent reduction on plaque density and microglial activation.

The technique used in this manuscript is novel and interesting. However, the manuscript is difficult to read due to the repetition of the methods section in the results. The data is properly discussed and the conclusion reached sustained by the results obtained. However, I have some concerns about this manuscript.

Major points:

1- The manuscript is too long. Although the three-dimensional histology must be explained in detail as it is the main point of this manuscript, the results section is impossible to understand. In all the subsections there is a long introduction repeating the information already stated in the methods section as well as, in some cases, a discussion of the results. This also occurs in the figure legends. This makes the manuscript tedious and distracts the interest of the reader. All this repeated information should be removed.

Minor points:

1- The authors state in the introduction that amyloid-PET has very low spatial resolution. Currently, the resolution of PET systems for rodents is under 1 mm, which in my opinion, is not low. Maybe the authors meant that the resolution is not enough to discriminate between plaques. This sentence should be rephrased.

2- The treatment is started before plaque deposition in this model. Thus, the effect observed is not a reduction on the plaques number but a prevention or deposition reduction.

3- Behavioral results are almost not discussed. For example, netoglitazone increased freezing behavior also in WT animals. This is not further discussed. In addition, there is no correlation of these results to the histology measurements. Which was the objective of performing these tests then?

4- The authors compare the 3D histology results to traditional 2D immunohistochemistry to show the superiority of the technique. However, there is no comparison to other quantitative technique, such as ELISA or even PET imaging, to validate the results.

7. PLOS authors have the option to publish the peer review history of their article (what does this mean? ). If published, this will include your full peer review and any attached files.

**Do you want your identity to be public for this peer review?** For information about this choice, including consent withdrawal, please see our Privacy Policy .

Reviewer #1: **Yes:** Andrew Octavian Sasmita

Reviewer #2: No

---

## [Author Response · Author response to Decision Letter 3]

17 Feb 2025

PLOS ONE Reviewer comments. 04.12.2024

Reviewer #1: The authors performed experiments to assess the modulatory effects of netoglitazone on Aβ plaque burden in vivo. The authors proposed that netoglitazone administration reduces Aβ plaque deposition and highlights the strength of volumetric assessment of Aβ plaque pathology in preclinical AD research. While the addition of information regarding asymmetrical plaque deposition is intriguing, there are still some concerns regarding the data presentation of the manuscript in its current form. See comments below:

1. Introduction

Regarding the amyloid cascade hypothesis, could the authors comment on the detrimental role that Aβ oligomers also play in inducing neuronal deficits? Although plaques seemingly exert physical detriments to the parenchyma and cells proximal to them, it remains debated whether an overabundance of oligomeric Aβ not bound in plaques would be more detrimental to neural functions. This could also explain why Aβ plaque deposition does not correlate well with cognitive decline, as it could be possible that plaques act as a reservoir for oligomeric Aβ from disrupting synaptic and other neural functions.

We thank the reviewer for this insightful observation. To address this point, we have added a section starting at line 67 of both the “Revised Manuscript with Track Changes” and the “Manuscript” in the introduction, which discusses the role of soluble Aβ oligomers in AD pathogenesis. Specifically, we highlighted how oligomers are highly neurotoxic. Additionally, we introduced the multifaceted role of plaques, acknowledging their contributions to localized tissue damage and neuroinflammation, as well as their potential to act as reservoirs for Aβ oligomers. This interplay between plaques and oligomers underscores their complexity as both drivers and modulators of AD pathology. References [6, 7] have been included to support this addition.

2. Methods:

a. Could the authors provide details on how the APPPS1 mice were bred? For instance, what is the sex of the transgene carrier for all breeding? A consistent reporting of breeding schemes would benefit readers and encourage transparency and reproducibility. Moreover, the usage of the term gender should be replaced with sex.

We thank the reviewer for this important suggestion. We have now clarified the breeding scheme used for the APPPS1 mice. Specifically, 68% of the transgene-carrying parents were male, while 32% were female. We added a sentence in line 130 of both the “Revised Manuscript with Track Changes” the “Manuscript” to explicitly describe the breeding scheme for clarity and reproducibility. Additionally, we have revised the manuscript to replace the term “gender” with “sex” to align with standard biological terminology.

b. Could the authors please indicate the different sex of animals (different symbols) used in graphs where pooling of male and female animals was performed? The authors noted in this revised manuscript that sex difference in freezing responses occur, but could the authors also comment on whether sex dimorphism in Aβ plaque load is evident between male and female APPPS1 mice? If sex dimorphism is evident, should the assessment of plaque load not be stratified further based on sex? In fact, the authors also openly stated in the discussion section that there is a disparity in PPARγ receptor expression between males and females.

We sincerely appreciate the reviewer’s thoughtful comments regarding the representation of sex differences in our study. In response, we have now indicated the sex of the animals using different symbols in Figure 1. Triangles represent females, while circles represent males.

Regarding Aβ plaque load, while sex differences in freezing responses and PPARγ receptor expression are acknowledged in our manuscript, our primary objective is to assess the overall therapeutic effects of the intervention on Aβ plaque load and behavioural outcomes in the APPPS1 cohort as a whole. In the case of conditioned fear assessment, we focused on male animals due to well-established sex differences in freezing responses, which could introduce behavioural variability unrelated to the intervention itself. However, Aβ pathology in the APPPS1 model is influenced by multiple factors, and while stratifying plaque load by sex could provide additional insights, our study was not specifically designed to investigate sex dimorphism in Aβ pathology. Introducing such an analysis could add complexity that extends beyond the scope of our primary research question. Nonetheless, we recognize the importance of sex as a biological variable and appreciate the reviewer’s perspective, which may be valuable for future investigations.

3. Results:

a. Figs 2 and 3 – The significant voxels and average values are very small and rather difficult to read. Is it possible to increase the size of these panels to improve legibility? Moreover, do early and late mean short-term (90 days) and long-term (180 days)? If so, the caption in the figures should be modified accordingly to improve clarity.

We thank the reviewer for their valuable feedback regarding Figures 2 and 3.

Panel Size: We acknowledge the difficulty in reading the significant voxels and average values due to their small size. While we attempted to enlarge the panels in Figures 2 and 3 in the revised manuscript, further increasing the size proved challenging without disrupting the layout. We have made adjustments to improve clarity as much as possible and hope this version is suitable.

Replacement of "Early" and "Late": In response to the reviewer’s comment, we have replaced the terms "early" and "late" with “low dose, 90 days” , “low dose, 180 days”, “high dose, 90 days”, and “high dose, 180 days” throughout the captions of Figures 2 and 3 for improved clarity.

b. Figs 2 – Could the authors elaborate on how significant voxels sometimes do not correspond to similar statistical differences between treated and untreated groups in the average count? Should conclusions not be drawn from the average count instead? Lastly, does the average count not correspond with the 2D analysis in Fig S4 which shows unchanged plaque load?

1. Significant Voxels vs. Average Count:

We appreciate the reviewer’s insightful question regarding the relationship between significant voxels and statistical differences observed in the average count and size of plaques. Voxel-based analysis provides a spatially resolved method for detecting regional differences in plaque distribution, allowing for localized statistical assessments that may not always align directly with aggregate measures. This discrepancy arises because voxel-wise comparisons account for spatial heterogeneity in plaque deposition, identifying localized effects that might be diluted when averaging across larger anatomical regions. In contrast, average plaque count comparisons across regions aggregate values over larger areas, potentially masking finer spatial patterns of significance that voxel-based analyses can detect. While the average count remains informative for overall cohort-level comparisons, voxel-based statistics provide additional granularity, highlighting spatially specific effects of treatment.

Thus, both measures provide complementary insights: average count offers a broader overview, while voxel-wise significance pinpoints localized treatment effects that might otherwise remain undetected.

Correspondence with Fig. S4:

We appreciate the reviewer’s question regarding the comparison between the 3D and 2D analyses. The difference in statistical outcomes between the two methods is primarily due to the resolution and comprehensiveness of the analyses.

In the 3D voxel-based analysis, significance is detected only in specific regions, while most other regions do not show statistical differences. This reflects a localized effect of the treatment rather than a global reduction in plaque load. In contrast, the 2D immunohistochemistry and immunofluorescence analysis provide a more limited sampling of brain regions, which may not capture these localized changes, leading to an overall non-significant result. Additionally, 3D analysis allows for a spatially resolved comparison across the entire brain volume, increasing sensitivity to detect regional effects that may not be apparent when averaging across larger anatomical structures in 2D. The statistical power of the 3D approach is therefore inherently different, as it considers voxel-level distributions rather than relying on sampled slices.

c. Fig 3 – Could the authors comment on why the graphs on the right in this figure depict the average plaque count again as opposed to the average plaque size? Are the average plaque counts per region plotted to again show differences caused by the treatment in a separate cohort of animals? From the table listing the number of experimental animals, it seems that the same mice were used to generate the data from Fig 2 and Fig 3, but the average count values from both figures do not match. Could the authors please clarify?

We thank the reviewer for pointing out these important issues and allowing us to clarify, and we sincerely apologize for any confusion caused.

Upon review, we realized that the legend for the graph in Fig. 3 labelled as "aggregate count" is incorrect and should instead read "mean aggregate size." We sincerely apologize for this oversight. The data depicted in the graphs on the right in Fig. 3 reflect the aggregate mean size per region, not the plaque count. This has been corrected in the revised figure legend to accurately represent the data. Regarding the cohort of animals, we confirm that the same mice were used to generate the data for both Fig. 2 and Fig. 3. The discrepancy in values between these figures is expected, as they represent different metrics: Fig. 2 shows plaque count, while Fig. 3 presents mean plaque size. We appreciate the reviewer’s careful attention to this detail and have revised the manuscript to ensure clarity and avoid any potential misunderstandings.

d. Fig 4A – In terms of data presentation, the white outline of the brain subregions seems to be too thick and hence distracting from the colours annotating the significant increases and decreases. If possible, could the authors please reduce the outline thickness or reduce the number of regions annotated to only show major subregions as in Figs 2 and 3?

We thank the reviewer for this valuable suggestion to improve the clarity of Fig. 4A. We have reduced the thickness of the white outlines in Fig. 4A to make the data presentation less distracting and ensure the colours annotating the significant increases and decreases are more prominent.

e. Fig 5 – In the main text, the authors described an abundance of DEGs associated with the high-dose treatment while DEGs are limited in mice that received low-dose treatment. In the figure, however, this is the opposite. Could the authors please clarify?

We appreciate the reviewer’s attention to detail and for pointing out this discrepancy. Upon review, we identified that the panels in Figure 5 were inadvertently switched. Panel A should correspond to the low-dose treatment, while Panel B represents the high-dose treatment. We sincerely apologize for the confusion caused by this labelling error in the figure legend. This correction ensures consistency between the figure and the main text, which states that the low-dose treatment resulted in a significant change, affecting 361 DEGs, whereas the high-dose treatment led to only a small change, with six DEGs being significantly altered.

4. Discussion:

a. The authors stated that the hippocampus and the cortex are prone to amyloid deposition. However, this statement seems misleading as the high plaque load in these regions can be directly linked to regions of Thy1-promoter activity (PMID: 12112467), which is the promoter used in the APPPS1-Jucker line. Hence, high plaque load in these regions is presumably a direct effect of transgene expression in a subset of neurons in said regions.

We thank the reviewer for this important observation. To address this, we have added a section starting from line 834 of the of the “Revised Manuscript with Track Changes” and line 773 of the “Manuscript” clarifying the factors contributing to regional differences in treatment efficacy and amyloid plaque distribution in the APPPS1-Jucker mouse model. This addition highlights the role of the Thy1-promoter in driving amyloid accumulation in the hippocampus and cortex, avoiding the potential misinterpretation that these regions are inherently prone to amyloid pathology. We believe this clarification improves the accuracy and transparency of the manuscript.

Reviewer #2: This study aims to interrogate the effect of netoglitazone in an animal model of Alzheimer's disease. To evaluate its effect, the authors perform behavioural tests and whole-brain immunohistochemistry. The authors found changes in behaviour and a dose-dependent reduction on plaque density and microglial activation.

The technique used in this manuscript is novel and interesting. However, the manuscript is difficult to read due to the repetition of the methods section in the results. The data is properly discussed and the conclusion reached sustained by the results obtained. However, I have some concerns about this manuscript.

Major points:

1- The manuscript is too long. Although the three-dimensional histology must be explained in detail as it is the main point of this manuscript, the results section is impossible to understand. In all the subsections there is a long introduction repeating the information already stated in the methods section as well as, in some cases, a discussion of the results. This also occurs in the figure legends. This makes the manuscript tedious and distracts the interest of the reader. All this repeated information should be removed.

We sincerely appreciate the reviewer’s feedback and understand the concern regarding the length and clarity of the results section. In response, we have carefully revised the manuscript to remove redundant information that was previously repeated from the methods section. We have streamlined the results section to focus on the key findings while ensuring that all necessary details for comprehension are retained.

Minor points:

1- The authors state in the introduction that amyloid-PET has very low spatial resolution. Currently, the resolution of PET systems for rodents is under 1 mm, which in my opinion, is not low. Maybe the authors meant that the resolution is not enough to discriminate between plaques. This sentence should be rephrased.

We thank the reviewer for this important clarification regarding the resolution of modern PET systems. We acknowledge that describing PET as having "very low spatial resolution" may be misleading, as current systems for rodents achieve sub-millimetre resolution, which is sufficient for regional analyses. However, our intended point was that this resolution remains inadequate for distinguishing individual amyloid plaques. To address this, we have rephrased the sentence starting from line 105 of both the “Revised Manuscript with Track Changes” and the “Manuscript” as follows: "While modern PET systems for rodents achieve sub-millimetre resolution, which is sufficient for regional analyses, their resolution remains insufficient to discriminate individual amyloid plaques." We believe this revision more accurately reflects the capabilities and limitations of PET systems and improves clarity. We appreciate the reviewer’s insightful comment, which has helped us refine our wording.

2- The treatment is started before plaque deposition in this model. Thus, the effect observed is not a reduction on the plaques number but a prevention or deposition reduction.

We thank the reviewer for raising this important point. In the first instance where the terminology for plaque decrease and plaque increase appears in the Results section, we have added a sentence starting from line 645 of the “Revised Manuscript with Track Changes” and line 611 of the “Manuscript” to clarify these terms: “Result of the reduction in the number of accumulated plaques, referred to as 'plaque decrease' from here on,” and similarly for increase:

---

## [Decision Letter · Decision Letter 3]

9 Mar 2025

PONE-D-24-31056R3Quantitative 3D histochemistry reveals region-specific amyloid-β reduction by the antidiabetic drug netoglitazonePLOS ONE

Dear Dr. Catto,

Thank you for submitting your manuscript to PLOS ONE. After careful consideration, we feel that it has merit but does not fully meet PLOS ONE’s publication criteria as it currently stands. Therefore, we invite you to submit a revised version of the manuscript that addresses the points raised during the review process.

We look forward to receiving your revised manuscript.

Kind regards,

Miriam A. Hickey, PhD

Academic Editor

PLOS ONE

**Journal Requirements:**

**Additional Editor Comments:**

Please respond to the minor revision requests by Reviewer #1.

Reviewers' comments:

Reviewer's Responses to Questions

**Comments to the Author**

1. If the authors have adequately addressed your comments raised in a previous round of review and you feel that this manuscript is now acceptable for publication, you may indicate that here to bypass the “Comments to the Author” section, enter your conflict of interest statement in the “Confidential to Editor” section, and submit your "Accept" recommendation.

Reviewer #1: All comments have been addressed

Reviewer #2: All comments have been addressed

2. Is the manuscript technically sound, and do the data support the conclusions?

Reviewer #1: Yes

Reviewer #2: Yes

3. Has the statistical analysis been performed appropriately and rigorously?

Reviewer #1: Yes

Reviewer #2: Yes

4. Have the authors made all data underlying the findings in their manuscript fully available?

Reviewer #1: Yes

Reviewer #2: Yes

5. Is the manuscript presented in an intelligible fashion and written in standard English?

Reviewer #1: Yes

Reviewer #2: Yes

6. Review Comments to the Author

**Reviewer #1:**  The authors have addressed the comments sufficiently. Perhaps two minor additions that could improve the discussion as the RNA sequencing data was not discussed/elaborated on as much:

1. Microglia are involved in plaque seeding via APOE (PMID: 39639016, 39419029), hence the changes in amyloidosis by netoglitazone could be mediated by microglial-mediated plaque formation. This could additionally explain the results of the high-dose netoglitazone observations as less microglia and plaques were observed. Additionally, since high-dose netoglitazone was shown to elevate CD68, the authors could argue using this data in the discussion to support their claim of elevated Aβ clearance.

2. The elevated Serpina3n is also interesting as it is a marker for disease-associated astrocytes (PMID: 32341542) and oligodendrocytes (PMID: 35760863) which could be relevant in other disease models where Aβ is not just produced by neurons or also in human cases as oligodendrocytes have recently been reported to contribute to Aβ production (PMID: 39103558).

**Reviewer #2:**  (No Response)

7. PLOS authors have the option to publish the peer review history of their article (what does this mean? ). If published, this will include your full peer review and any attached files.

**Do you want your identity to be public for this peer review?** For information about this choice, including consent withdrawal, please see our Privacy Policy .

Reviewer #1: No

Reviewer #2: No

---

## [Author Response · Author response to Decision Letter 4]

24 Mar 2025

PLOS ONE Reviewer comments from 10.03.2025

Reviewer #1: The authors have addressed the comments sufficiently. Perhaps two minor additions that could improve the discussion as the RNA sequencing data was not discussed/elaborated on as much:

1. Microglia are involved in plaque seeding via APOE (PMID: 39639016, 39419029), hence the changes in amyloidosis by netoglitazone could be mediated by microglial-mediated plaque formation. This could additionally explain the results of the high-dose netoglitazone observations as less microglia and plaques were observed. Additionally, since high-dose netoglitazone was shown to elevate CD68, the authors could argue using this data in the discussion to support their claim of elevated Aβ clearance.

2. The elevated Serpina3n is also interesting as it is a marker for disease-associated astrocytes (PMID: 32341542) and oligodendrocytes (PMID: 35760863) which could be relevant in other disease models where Aβ is not just produced by neurons or also in human cases as oligodendrocytes have recently been reported to contribute to Aβ production (PMID: 39103558).

Answer:

We sincerely appreciate the reviewer’s insightful comments. In response, we have incorporated the suggested additions beginning at line 883 (in the Manuscript with track changes) and 821 (in the Manuscript) of the discussion, expanding on microglial involvement in plaque formation, the role of CD68 in Aβ clearance, and the relevance of Serpina3n to astrocytes, oligodendrocytes, and Aβ production. These valuable suggestions have further strengthened our manuscript, and we are grateful for the reviewer’s constructive feedback.

Response to Journal Requirements

We have thoroughly reviewed and updated the reference list to ensure accuracy, completeness, and clarity. The following revisions have been made:

Added:

• References 106-112 to support the recently incorporated section in the discussion.

• Nishiyama, K., et al., Presenilin 1 mRNA expression in hippocampi of sporadic Alzheimer's disease patients. Neurosci Res, 1996. 26(1): p. 75-8, to enhance clarity.

• Zhang, Y., et al., Amyloid β-based therapy for Alzheimer’s disease: challenges, successes, and future. Signal Transduction and Targeted Therapy, 2023. 8(1): p. 248, to strengthen the discussion.

Removed:

• The Human Protein Atlas, which was mistakenly included and was not necessary in the context. We apologize for the oversight.

• Schalbetter, S.M., et al., Adolescence is a sensitive period for prefrontal microglia to act on cognitive development. Science Advances, 2022, as it was considered non-essential.

Replaced:

• Rafii, M.S., & Aisen, P.S., Brain region-specific pharmacodynamics. Alzheimer's & Dementia, 2009 → Replaced with Srinivas, N., K. Maffuid, and A.D.M. Kashuba, Clinical Pharmacokinetics and Pharmacodynamics of Drugs in the Central Nervous System. Clin Pharmacokinet, 2018. 57(9): p. 1059-1074 for a more recent and relevant reference.

• HABCHI, J.Y., et al., THERAPY FOR PROTEIN MISFOLDING DISEASE, W.T. LIMITED, 2019 → Replaced with Habchi, J., et al., Systematic development of small molecules to inhibit specific microscopic steps of Aβ42 aggregation in Alzheimer’s disease. PNAS, 2017. 114(2): p. E200-E208 for improved citation accuracy.

Resolved: All reference repetitions have been resolved.

We appreciate PlosOne guidance in refining our reference list and remain open to any further adjustments as needed.

---

## [Decision Letter · Decision Letter 4]

30 Mar 2025

Quantitative 3D histochemistry reveals region-specific amyloid-β reduction by the antidiabetic drug netoglitazone

PONE-D-24-31056R4

Dear Dr. Catto,

We’re pleased to inform you that your manuscript has been judged scientifically suitable for publication and will be formally accepted for publication once it meets all outstanding technical requirements.

Kind regards,

Miriam Ann Hickey, PhD

Academic Editor

PLOS ONE

Additional Editor Comments (optional):

The authors have now addressed all comments satisfactorily.

Reviewers' comments:

Reviewer's Responses to Questions

**Comments to the Author**

1. If the authors have adequately addressed your comments raised in a previous round of review and you feel that this manuscript is now acceptable for publication, you may indicate that here to bypass the “Comments to the Author” section, enter your conflict of interest statement in the “Confidential to Editor” section, and submit your "Accept" recommendation.

Reviewer #1: (No Response)

2. Is the manuscript technically sound, and do the data support the conclusions?

Reviewer #1: Yes

3. Has the statistical analysis been performed appropriately and rigorously?

Reviewer #1: Yes

4. Have the authors made all data underlying the findings in their manuscript fully available?

Reviewer #1: Yes

5. Is the manuscript presented in an intelligible fashion and written in standard English?

Reviewer #1: Yes

6. Review Comments to the Author

Reviewer #1: (No Response)

7. PLOS authors have the option to publish the peer review history of their article (what does this mean? ). If published, this will include your full peer review and any attached files.

**Do you want your identity to be public for this peer review?** For information about this choice, including consent withdrawal, please see our Privacy Policy .

Reviewer #1: No

---

## [Editor Report · Acceptance letter]

PONE-D-24-31056R4

PLOS ONE

Dear Dr. Catto,

I'm pleased to inform you that your manuscript has been deemed suitable for publication in PLOS ONE. Congratulations! Your manuscript is now being handed over to our production team.

Kind regards,

on behalf of

Dr. Miriam Ann Hickey

Academic Editor

PLOS ONE